# Modeling submerged biofouled microplastics and their vertical trajectories

Reint Fischer[1*], Delphine Lobelle[1*], Merel Kooi[2], Albert Koelmans[2], Victor Onink[1,3,4],
Charlotte Laufkötter[3,4], Linda Amaral-Zettler[5], Andrew Yool[6], and Erik van Sebille[1,7]

[1]Institute for Marine and Atmospheric Research, Utrecht University, Utrecht, Netherlands
[2]Aquatic Ecology and Water Quality Management Group, Department of Environmental Sciences, Wageningen University
[3]Climate and Environmental Physics, Physics Institute, University of Bern, 3012 Bern, Switzerland
[4]Oeschger Centre for Climate Change Research, University of Bern, 3012 Bern, Switzerland
[5]Royal Netherlands Institute for Sea Research, Netherlands
[6]National Oceanography Centre, Southampton, UK
[7]Centre for Complex Systems Studies, Utrecht University, Utrecht, Netherlands
[*]These authors contributed equally to this work.

**Correspondence:** DELPHINE LOBELLE d.m.a.lobelle@uu.nl

**Abstract.** The fate of (micro)plastic particles in the open ocean is controlled by biological and physical processes. Here, we model the effects of biofouling on the subsurface vertical distribution of spherical, virtual plastic particles with radii of 0.01–1 mm. The biological specifications include the attachment, growth and loss of algae on particles. The physical specifications include four vertical velocity terms: advection, wind-driven mixing, tidally-induced mixing, and the sinking velocity of the biofouled particle. We track 10,000 particles for one year in three different regions with distinct biological and physical properties: the low productivity region of the North Pacific Subtropical Gyre, the high productivity region of the Equatorial Pacific and the high mixing region of the Southern Ocean. The growth of biofilm mass in the euphotic zone and loss of mass below the euphotic zone result in the oscillatory behaviour of particles, where the larger (0.1–1.0 mm) particles have much shorter average oscillation lengths (<10 days; 90th percentile) than the smaller (0.01–0.1 mm) particles (up to 130 days; 90th percentile). A subsurface maximum particle concentration occurs just below the mixed layer depth (around 30 m) in the Equatorial Pacific, which is most pronounced for larger particles (0.1–1.0 mm). This occurs since particles become neutrally buoyant when the processes affecting the settling velocity of a particle and the seawater's vertical movement are in equilibrium. Seasonal effects in the subtropical gyre result in particles sinking below the mixed layer depth only during spring blooms, but otherwise remaining within the mixed layer. The strong winds and deepest average mixed layer depth in the Southern Ocean (400 m) result in the deepest redistribution of particles (>5000 m). Our results show that the vertical movement of particles is mainly affected by physical (wind-induced mixing) processes within the mixed layer and biological (biofilm) dynamics below the mixed layer. Furthermore, positively buoyant particles with radii of 0.01–1.0 mm can sink far below the euphotic zone and mixed layer in regions with high near-surface mixing or high biological activity. This work can easily be coupled to other models to simulate open-ocean biofouling dynamics, in order to reach a better understanding of where ocean (micro)plastic ends up.

# 1 Introduction

Observations have shown that plastic ends up everywhere in the ocean, from the Arctic (Cózar et al., 2017), to the Mariana Trench (Chiba et al., 2018; Peng et al., 2018). Two key questions that many studies address are: how much plastic is found in each ocean reservoir and how does it get there? To tackle the first question, studies using the 2010 global Jambeck et al. (2015) estimate of plastic entering the ocean from the coasts (4.8–12.7 million tons), suggest that approximately 1% is at the sea surface (Eriksen et al., 2014; Sebille et al., 2015) and 67-77% ends up on beaches or in coastal waters, up to 10 km offshore (Lebreton and Andrady, 2019; Onink et al., 2021). Although a recent study by Weiss et al. (2021) suggests that Jambeck et al. (2015) overestimates plastic fluxes from rivers to oceans by two to three orders of magnitude, this would not affect the fraction of the total that ends up close to the coasts from Onink et al. (2021), for example. Following these approximations, around 20-30% of ocean plastic debris is unaccounted for and could either be in the water column or on the seafloor. The focus of this study is therefore to explore the processes affecting the vertical distribution of ocean plastic.

Logistical constraints result in limited (and sometimes only shallow and coastal) subsurface observations of plastic concentration (e.g. Reisser et al., 2015; Kanhai et al., 2017; Dai et al., 2018; Pabortsava and Lampitt, 2020; Kukulka et al., 2012). However, a few recent regional open ocean samples have been obtained. Observations down to 2000 m traversing the North Pacific Subtropical Gyre (Egger et al., 2020) show that the highest concentration of 0.5 to 50 mm-sized particles is at the surface and then a power-law decline occurs with depth (with surface concentrations up to four orders of magnitude larger than at deeper depths). In both the Monterey Bay, 25 km offshore (Choy et al., 2019), and in three regions in the East Indian Ocean (Li et al., 2020), a subsurface maximum concentration of 0.1 to 5 mm-sized particles has been observed around 200 m (just below the average mixed layer depth in the Monterey Bay). These studies show that vertical plastic concentration profiles vary spatially, and we therefore investigate the physical and biological mechanisms that affect such variation.

Among the physical processes controlling the sinking of buoyant plastic particles (Maxey and Riley, 1983; Sebille et al., 2020), large-scale advection has been shown to play a role. For example, Lobelle et al. (2021) show that in downwelling subtropical gyres, modeled $1 \mu$m particles remain at the surface without vertical advection, and can start sinking within 60 days with advection. Finer-scale physical processes have also been shown to affect the vertical transport of plastic. Model studies simulating the effects of turbulent mixing driven by wind and buoyancy loss demonstrate that the upward flux of smaller buoyant particles ($<$1mm) could be balanced by downward mixing and particles could be vertically redistributed within the mixed layer (Kukulka et al., 2012; Enders et al., 2015). In some regions, vertical mixing from internal tides is shown to affect the dispersal of small particles (such as larvae in deep-sea hydrothermal vents; Vic et al., 2018). Furthermore, de Lavergne et al. (2020) have shown that mixing in the interior is at first order governed by mixing driven by internal tides. Previous work by Mountford and Maqueda (2019, 2021) has also shown the importance of interior diapycnal mixing for the dispersal of plastic in the ocean. We therefore include diapycnal mixing (tidally-induced) in this study both to test whether it can impact near-surface particle displacement as well as to provide full-depth mixing dynamics (and not solely within the mixed layer).

Floating debris smaller than 5 mm can also sink below the sea surface as a result of biological processes, such as getting encapsulated in fecal pellets and marine snow (Cole et al., 2016; Kvale et al., 2020) or biofouling (Ye and Andrady, 1991;

Bravo et al., 2011; Lobelle and Cunliffe, 2011; Fazey and Ryan, 2016; Kooi et al., 2017; Amaral-Zettler et al., 2021b), and here we focus on the latter. Our study is based on Kooi et al. (2017) who present an idealised 1D, depth-dependent biofouling model using fixed water property profiles in a quiescent ocean. As previously hypothesised by Ye and Andrady (1991), Kooi et al. (2017) suggest that particles can sink due to algal attachment and growth and can rise due to defouling or biofilm death below the euphotic zone; they oscillate around the euphotic zone depth. Lobelle et al. (2021) then extended the Kooi et al. (2017) model to investigate the global initial sinking characteristics of particles in a temporally and spatially varying framework, while including 3D advection. This study builds on the work in Lobelle et al. (2021), with some key changes and improvements. Firstly, we include small-scale vertical turbulence which is important for the vertical transport of smaller plastic particles within the mixed layer (Kukulka et al., 2012). We also improve the biofilm loss terms by dynamically computing grazing as opposed to using a constant rate, and account for other biofilm losses, from viruses, for example. Lastly, we focus on three open-ocean regions to highlight effects of contrasting physical and biological seawater properties, i.e. high and low algal concentrations and high and low wind-induced mixing.

The aim of this study is to address the knowledge gap surrounding the spatial distribution of vertical processes affecting floating (micro)plastic sinking from the ocean surface (Sebille et al., 2020). We explore the vertical distribution and oscillations of particles with a radius between 0.01 and 1 mm that have undergone initial biofouling in the open ocean, while representing the physical and biological dynamics as realistically as possible.

## 2 Method

### 2.1 Forcing data and domain

We track virtual microplastic particles using the Parcels 2.2.2 Lagrangian framework (Delandmeter and van Sebille, 2019). The NEMO-MEDUSA-2.0 ORCA0083-N06 output (Yool et al., 2013), hereafter NEMO-MEDUSA, is used for the hydrodynamic and biological data (available from http://opendap4gws.jasmin.ac.uk/thredds/nemo/root/catalog.html). The resolution in NEMO-MEDUSA is 1/12° horizontally, 75 levels vertically and five days temporally. We select fifteen months of data (from 1 October 2003 to 31 December 2004) to allow for three months of initial model spin-up and then one full year of simulations to analyse seasonality. We chose 2004 in order to remain consistent with the year chosen in Lobelle et al. (2021), that represents a 'typical year' for the sinking characteristics of particles. We have also repeated the analysis using two years plus three months to verify the seasonality of the one-year results (Fig. B1). The spin-up time allows particles to be initially biofouled, since microorganisms can colonise plastic and increase its density enough to make it sink within about 6 weeks (Lobelle and Cunliffe, 2011; Fazey and Ryan, 2016; Kaiser et al., 2017). We are therefore simulating more realistic fouled particles in the ocean, as opposed to the previous model version in Lobelle et al. (2021) using clean, pristine particles.

We focus on three regions in this study, which have different physical and biological profiles: (1) the Equatorial Pacific (EqPac) from -4.5° to 4.5° N and 139° to 148° W, (2) the North Pacific Subtropical Gyre (NPSG) from 23° to 32° N and 134° to 143° W and (3) the Southern Ocean (SO) from -53° to -62° N and 106° to 115° W (Fig. 1). The annually and horizontally averaged profiles of vertical mixing, diatom concentration and primary productivity reveal the differences between these three

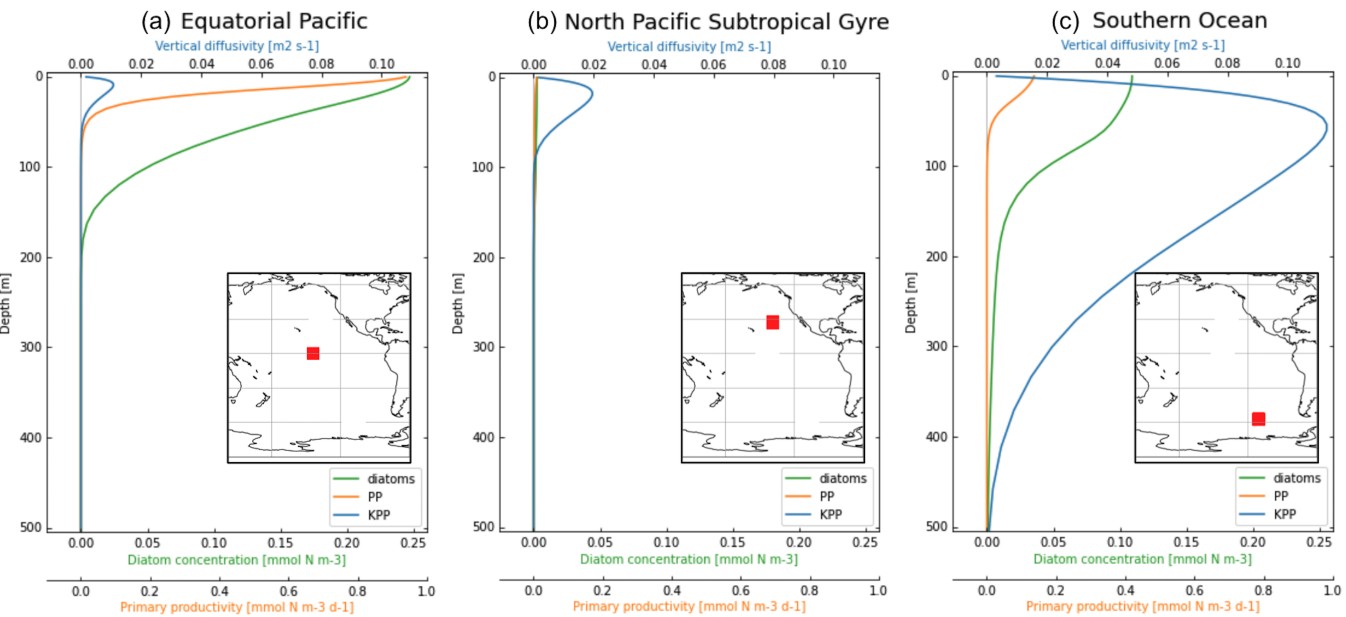

**Figure 1.** Vertical profiles averaged over 2004 of KPP vertical mixing in blue (m$^2$ s$^{-1}$; top x-axis), diatom concentration in green (mmol N m$^{-3}$; bottom x-axis) and primary productivity in orange (mmol N m$^{-3}$ s$^{-1}$; bottom x-axis). The profiles are also averaged horizontally in each 10x10° location: (a) Equatorial Pacific, (b) North Pacific Subtropical Gyre and (c) Southern Ocean, with the exact release locations shown by the red boxes in the embedded maps.

regions (see Sect. 2.2 for how we define vertical mixing and Sect. 2.4 for the biological variables). The EqPac has the lowest maximum annual average vertical mixing of the three regions (0.01 m$^2$ s$^{-1}$ at 15 m; Fig. 1a), however it has the highest diatom concentration and primary productivity (of almost 0.25 mmol N m$^{-3}$ and 1.0 mmol N m$^{-3}$ d$^{-1}$ at the surface, respectively).

Diatoms are found in this region down to approximately 200 m. The vertical mixing profile in the NPSG reaches a slightly higher maximum than in the EqPac (0.02 m$^2$ s$^{-1}$; Fig. 1b). However, due to the warm, stratified surface waters with low nutrient levels, diatom concentrations and primary productivity are very low in these oligotrophic waters. Lastly, the average vertical mixing profile in the SO reaches almost an order of magnitude higher than the other two regions (0.12 m$^2$ s$^{-1}$) and extends to 500 m deep (Fig. 1c).

In each of the three regions, we initiate a total of 100 release locations on a 1° x 1° grid at $z = 0.6$ m (the approximate midpoint depth of the surface grid cell in NEMO). Within the Parcels Lagrangian framework, a C-grid interpolation scheme is used for temporal and spatial interpolation of the fields (Delandmeter and van Sebille, 2019). The three-dimensional fourth-order Runge-Kutta method is used with an integration time step of 60 seconds, and the 3D position and biofouling state of each particle is stored every 12 hours. We tested the sensitivity of our results using an integration time step of 30 seconds and the

results did not change (though the simulation time was longer); with a longer integration time step (e.g. 1 hour) we lost some information for shorter oscillation frequencies (e.g. for 1 mm particles).

The particles in our simulations represent spherical plastic with a radius between 0.01 and 1 mm. The upper limit is the same as in Lobelle et al. (2021) and the lower limit is to comply with the lowest order of magnitude on which diatoms have been observed to attach (Amaral-Zettler et al., 2021a, b). We use 25 size bins within this range, releasing 4 identical particles per bin at each release location (that have different trajectories due to mixing). Since we have 100 release locations in each region of study, we simulate 400 particles per bin size per region, and 10,000 particles in total per region.

We also assign an initial density to the virtual particles. Following results in Lobelle et al. (2021) showing that sinking characteristics of biofouled plastic particles with initial densities of 30 and 920 kg m$^{-3}$ show minor differences, we focus on one density here, 920 kg m$^{-3}$ (representative of low-density polyethylene; one of the most commonly produced plastic polymers). We have also run two sensitivity analyses for particles with a density of 30 kg m$^{-3}$ (representative of expandable polystyrene) and 1020 kg m$^{-3}$ (representative of rigid polyamide). We found that the only difference to 920 kg m$^{-3}$ results is that in the oligotrophic NPSG, 30 kg m$^{-3}$ particles of the largest size class remain at the surface instead of sinking to the base of the mixed layer (Fig. C1b). For the 1020 kg m$^{-3}$ simulations in the NPSG (Fig. C2b and e), the majority of the larger particles mix completely to the base of the MLD (as opposed to 920 kg m$^{-3}$ particles mostly staying close to the sea surface). The smaller 1020 kg m$^{-3}$ particles on average resurface slower after being mixed down to 200 m in spring (as opposed to 920 kg m$^{-3}$ particles that quickly resurface). Particles representing other sizes in other regions with a density of 30 and 1020 kg m$^{-3}$ produce very similar results to the 920 kg m$^{-3}$ particles. The particles in our simulations are spherical and we assume that they do not fragment, change shape or change density throughout the simulations.

The physical data used from NEMO-MEDUSA are: potential temperature [°C], salinity [psu], vertical velocity [m s$^{-1}$], wind stress [N m$-3$], wind speed [m s$^{-1}$; at 10 m], the mixed layer depth (MLD in m; defined as the depth at which potential density changes by 0.01 kg m$^{-3}$ relative to the surface) and the euphotic zone depth [EZD in m]. The wind stress, wind speed and mixed layer depth are used to compute the wind-induced mixing in the upper ocean, described in Sect. 2.2. As the aim of our study is to analyse the regional signal, we run all simulations in the vertical dimension only, however we have also tested the sensitivity of our results to 3D advection (Appendix E), where the horizontal velocities from NEMO-MEDUSA are also used for these simulations. The biological data used are: concentrations of diatom phytoplankton, microzooplankton and mesozooplankton [each in mmol N m$^{-3}$] and total primary productivity [mmol N m$^{-3}$ d$^{-1}$]. These data are used to define the biofouling dynamics, described in Sect. 2.4. The biogeochemical performance of NEMO-MEDUSA has previously been extensively validated at low resolution in the studies of Yool et al. (2013) and Yool et al. (2021), with traceability at higher resolution demonstrated in Yool et al. (2015).

## 2.2 Physical dynamics

There are three components that make up the physical dynamics included in this study: vertical advection (from NEMO-MEDUSA), computed wind-induced vertical mixing and computed tidally-induced diapycnal vertical mixing. Wind-driven turbulence can play an important role in the vertical concentration profiles of buoyant particles (Kukulka et al., 2012) and

**Table 1.** The comparison between the modeled physical and biofilm dynamic specifications, progressing from the Kooi et al. (2017) model to the Lobelle et al. (2021) model and to the model in this study.

| Models | | Kooi et al. 2017 | Lobelle et al. 2021 | Current study |
|---|---|---|---|---|
| **Physics** | Horizontal advection | | x | |
| | Vertical advection | | x | x |
| | Parameterised vertical mixing | | | x |
| **Biofilm** | Gain via collisions | x | x | x |
| | Gain via growth | x | x | x |
| | Loss via respiration | x | x | x |
| | Loss via explicit grazing | | | x |
| | Loss via viral lysis | | | x |

therefore its inclusion is one of the novelties of this study (Table 1). Since we do not have access to the diffusivity profiles from NEMO-MEDUSA, we follow the approach from Onink et al. (*subm.*) to model turbulent stochastic transport in the surface mixed layer using a Markov-0 random walk model. The amount of turbulence in the surface mixed layer is computed using the K-profile parametrisation (KPP) (Large et al., 1994; Boufadel et al., 2020), $K_z$, given by:

$$K_z = \left(\frac{\kappa u_{*w}}{\phi}\theta\right)(|z| + z_0)\left(1 - \frac{|z|}{MLD}\right)^2,  \tag{1}$$

where $\kappa = 0.4$ is the Von Kármán constant, $u_{*w}$ is the friction velocity of the seawater's surface [m s$^{-1}$], $\phi = 0.9$ is the "stability function" of the Monin-Obukov boundary layer theory, $\theta = 1$ is a Langmuir circulation enhancement factor (which is 1 when Langmuir cells are negligible, as is the assumption here), $z_0$ is the roughness scale of turbulence [m] (defined below in Eq. (2)), and $MLD$ is the mixed layer depth [m], provided by NEMO-MEDUSA. Note that neglecting Langmuir enhanced mixing in the Southern Ocean might not be realistic (Li et al., 2016), however for simplicity and standardisation, we keep this assumption constant in all regions. The roughness scale, $z_0$ [m], depends on the wind speed and the wave age (Zhao and Li, 2019). Assuming a constant wave age $\beta_* == c_p/u_{*a} = 35$ for a fully developed wave state, like in Kukulka et al. (2012), where $c_p$ is the wave phase speed [-] and $u_{*a}$ is the friction velocity of air [m s$^{-1}$], the roughness scale based on Zhao and Li (2019) is:

$$z_0 = 3.5153 \times 10^{-5}\left(\frac{\beta_* u_{*a}}{u_{10}}\right)^{-0.42}u_{10}^2/g,  \tag{2}$$

where $u_{10}$ is the 10 m wind speed [m s$^{-1}$] and $g = 9.81$ m s$^{-2}$ is the acceleration due to gravity. We use a local form of the KPP profile, where we neglect non-local terms for simplicity. Our boundary condition at the surface is set such that if a particle is about to cross the surface, we set its depth to the NEMO-MEDUSA surface depth (0.6 m). Onink et al. (*subm.*) show that for positively buoyant particles, 1D vertical profiles estimated with this Markov-0 approach match reasonably well with observations, where increased wind stress results in particles being mixed to greater depths and reduced particle rise velocities.

The vertical stochastic velocity perturbation due to turbulent mixing, $w_m$, according to Gräwe et al. (2012) and solved using an Euler-Maruyama scheme (Maruyama, 1955), is given by:

$$w_m(t) = \partial_z K_z + \frac{1}{dt}\sqrt{2K_z}dW, \tag{3}$$

$$Z(t+\Delta t) = Z(t) + \big(w_s(t) + w_a(t) + w_m(t)\big)\Delta t, \tag{4}$$

where $\partial_z K_z = \partial K_z/\partial z$, $\Delta t$ is the integration time step, $dW$ is the Wiener increment with a mean of 0 and standard deviation, $\sigma = \sqrt{\Delta t}$, $w_a$ is the vertical advection [m s$^{-1}$] and $w_s$ is the sinking velocity of the particle [m s$^{-1}$] defined in Eq. (5).

Since KPP only estimates turbulent mixing above the MLD, we also include a background full-depth vertical diapycnal mixing induced by internal tides (see de Lavergne et al. (2020) for the detailed methodology). The global tidal mixing maps they provide which we use to estimate tidally formed $K_z$ are available from: https://www.seanoe.org/data/00619/73082/.

## 2.3 Particle settling velocity

The Lobelle et al. (2021) biofouling model transformed the Kooi et al. (2017) 1D vertical model into full 3D (Table 1). The core of the Kooi et al. (2017) model remains the same in our study, namely that the settling velocity of a particle depends on two factors: 1) the density difference between the biofouled plastic particle and the surrounding seawater and 2) the size and density of the particle:

$$w_s(x,y,z,t) = -\left(\frac{\rho_{tot} - \rho_{sw}}{\rho_{sw}}g\omega_* \upsilon_{sw}\right)^{1/3}, \tag{5}$$

where $\rho_{tot}$ is the total density of the particle plus attached algae [kg m$^{-3}$], $\rho_{sw}$ is the ambient seawater density [kg m$^{-3}$] derived from NEMO-MEDUSA's temperature and salinity fields that vary in 3D time and space, using the TEOS-10 standard equation of state (see Roquet et al., 2015; McDougall et al., 2003), $\omega_*$ is the dimensionless settling velocity and $\upsilon_{sw}$ is the kinematic viscosity of the seawater [m$^2$ s$^{-1}$]. The settling velocity, $w_s$ [m s$^{-1}$], can therefore be computed as a function of the three spatial directions and time ($x$, $y$, $z$ and $t$). The supplementary material of Lobelle et al. (2021) describes the equations behind each term in Eq.( 5). Also, here the kinematic viscosity has been computed dynamically in 3D space and time as opposed to using the same profile as defined in Kooi et al. (2017), though the spatiotemporal variations are so small that this modification has a minor impact on the results (Chen et al., 1973).

## 2.4 Biofilm dynamics: gain and loss terms

In this study we assume that the biofilm solely consists of diatoms. While the model described in Lobelle et al. (2021) characterised biofilms as being comprised of diatoms and non-diatoms, Amaral-Zettler et al. (2020) did not observe the species of phytoplankton that make up the bulk of the non-diatoms in their observational study of biofilms. Therefore we choose to limit the species to diatoms, which we know to be found on ocean plastic. The attached biofilm, $\frac{dA}{dt}$ affects the total volume and density of the particle + biofilm (Eq. (5)–(10) in Kooi et al., 2017), which determines the dimensionless settling velocity $\omega_*$ in

Eq. (5) above. In this study, we use two gain terms and three loss terms to define the algal biofilm dynamics:

$$\frac{dA}{dt} = G_{coll} + G_{grow} - L_{graze} - L_{resp} - L_{nonlin}. \tag{6}$$

The two gain terms are identical to the Lobelle et al. (2021) model (Table 1). As such, $G_{coll} = \frac{A_A \beta_A}{\theta_{pl}}$ [no. m$^{-2}$ s$^{-1}$] represents a particle's collision with and colonisation by algae; $\beta_A$ is the collision rate [m$^3$ s$-1$], $A_A$ is the planktonic algal concentration [no. m$^{-3}$] and $\theta_{pl}$ is the surface area of the spherical plastic [m$^2$] (see Eq. (S15)–(S17) in Lobelle et al., 2021). Note that the term 'planktonic' algae is used in this study to refer to the algae present in seawater and 'attached' to refer to the algae present in the biofilm. The growth of the attached algal cells is $G_{grow} = \mu_A A$ [no.m$^{-3}$ s$^{-1}$], where $\mu_A$ is computed from the 3D total

primary productivity (TPP3) output from NEMO-MEDUSA (mmol N m$^{-3}$ d$^{-1}$). Since TPP3 is the total primary productivity of both diatoms and non-diatoms, and the productivity of diatoms alone is not available, we assume that the rate is the same for both phytoplankton. We divide TPP3 by the total planktonic diatom + non-diatom concentration, and use that rate to multiply by the total number of attached diatoms ($A$). Therefore, after converting TPP3 to an algal growth rate by multiplying by the atomic weight of nitrogen (14.007 g) and then using the median nitrogen to algal cell ratio (356.04 $\times$ 10$^9$; Lobelle et al.,

2021), we divide by the total diatom + non-diatom concentration to get a rate [d$^{-1}$]. That rate is then multiplied by the attached diatom algal biofilm to define $G_{grow}$. We also set a maximum growth rate of 1.85 d$^{-1}$ (Bernard and Rémond, 2012), following the Kooi et al. (2017) model. We define the biofilm density as 1170 kg m$^{-3}$, following results from Amaral-Zettler et al. (2021b); though it may seem counter-intuitive that organisms denser than seawater are found at the surface, they are retained there due to upwelling or mixing. Also, they can transfer from other floating particles (seaweed, feathers, marine snow, etc) to

plastic at the surface. We also tested the use of a denser biofilm (1388 kg m$^{-3}$ as in Kooi et al., 2017) and our results did not change (not shown).

There are only two loss terms in the Lobelle et al. (2021) model including temperature-dependent respiration and a fixed constant for grazing (following the Kooi et al. (2017) model). Here, we aim to improve the biofilm dynamics and use the underlying equations from NEMO-MEDUSA (Yool et al., 2013) to compute spatially and temporally varying grazing of

205 diatoms in biofilms as well as adding another term, nonlinear losses, which include viral lysis (Table 1). The grazing of diatoms by mesozooplankton is available from NEMO-MEDUSA, however, only as a depth-integrated variable (in mmol N m$^{-2}$ s$^{-1}$). We therefore recompute 3D depth-dependent grazing (in mmol N m$^{-3}$ s$^{-1}$) dynamically using Eq. (54) from Yool et al. (2013):

$$L_{graze} = \frac{g_m \cdot p_{mPd} \cdot Pd^2 \cdot Zm}{k_m^2 + F_m}. \tag{7}$$

Here, $g_m$ is the maximum zooplankton grazing rate (0.5 d$^{-1}$), $p_{mPd}$ is the dimensionless mesozooplankton preference for diatoms (0.35), $Pd$ is the diatom concentration from NEMO-MEDUSA [mmol N m$^{-3}$], $Z_m$ is the mesozooplankton concentration from NEMO-MEDUSA [mmol N m$^{-3}$], $k_m$ is the zooplankton grazing half-saturation constant (0.3 mmol N m$^{-3}$) and $F_m$ is a composite term of mesozooplankton preference for total available food (including non-diatoms, diatoms, microzooplankton and detritus; see Eq. (55) in Yool et al., 2013). These equations are for the grazing of diatom phytoplankton in

seawater, and the biofilm we model is in units of number of algal cells, following the Kooi et al. (2017) model dynamics, so

the same N:algal cell conversion as described above is used. We then divide by the number of attached algal cells in Eq. (6) to get a grazing rate. This means that we assume that the mesozooplankton grazing rate is the same for the planktonic algae as for the attached algae on microplastic at a specific point in time and depth.

The loss rate via respiration remains as in the Kooi et al. (2017) model and is therefore:

$$L_{resp} = Q_{10}^{(T-20)/10} R_{20} A, \tag{8}$$

where $R_{20} A = 0.1$ d$^{-1}$ with the coefficient, $Q_{10} = 2$, which represents how much the respiration rate increases by every $10°$ C increase in temperature; where $T$ is the NEMO-MEDUSA temperature field [$°$ C] that varies in 3D time and space (see Fig. F1 for the graphical relationship of the respiration rate and seawater temperature).

The final loss term represents processes that depend on the abundance of diatoms, such as diseases (including viral lysis). This term is represented using a saturating hyperbolic function defined in Eq. (72) of Yool et al. (2013):

$$L_{nonlin} = \lambda \frac{Pd}{k_{Pd} + Pd} Pd, \tag{9}$$

where $\lambda$ is a nonlinear maximum loss rate of 0.1 d$^{-1}$, and $k_{Pd}$ is the loss half-saturation constant (0.5 mmol N m$^{-3}$). As with the grazing above, these nonlinear losses are determined relative to the abundance of diatoms in seawater. This loss rate is then applied to the number of algae attached to the particle.

These biofilm gain and loss terms result in an oscillatory behaviour of the particles due to the biofilm's gain causing an increase in overall density and sinking followed by the biofilm's loss and decrease in density which leads to rising. We also propose an alternative scenario where algal cell walls remain attached in the dark (see the full description in Appendix D).

One of the key assumptions we make in this study is that the biofilm only consists of phytoplankton (diatoms), since this is the data available in NEMO-MEDUSA. Understanding the composition of the biofilm community (or plastisphere) is important to accurately model the effects of biofilm dynamics on the vertical motion of particles. Recent work in the Mediterranean and the North Sea's coastal waters (Amaral-Zettler et al., 2021a, b) observed that for a spherical polyethylene particle with a radius of $30\mu$m or smaller, small single-celled organisms (including bacteria) need to be considered. We therefore focus on simulating plastic particles that are large enough for diatoms to attach. The limitations of this assumption are further explored in Sect. 3.4.

## 3 Results and discussion

### 3.1 The vertical distribution of particles

Our work supports the findings from previous studies that floating particles can sink in the open ocean due to biofouling. The sinking behaviour (and vertical distribution) of particles varies for different locations and particle sizes (Kaiser et al., 2017; Choy et al., 2019; Lobelle et al., 2021). We compare vertical one-year trajectories in three regions (Equatorial Pacific; EqPac, North Pacific Subtropical Gyre; NPSG, and Southern Ocean; SO) and for two particle radius size classes (0.01–0.1 mm and 0.1–1.0 mm; Fig. 2). We reiterate that our simulations only include vertical motion (advection and mixing) in order to isolate and contrast specific biological and physical factors that affect vertical particle displacement. We have also tested the effects

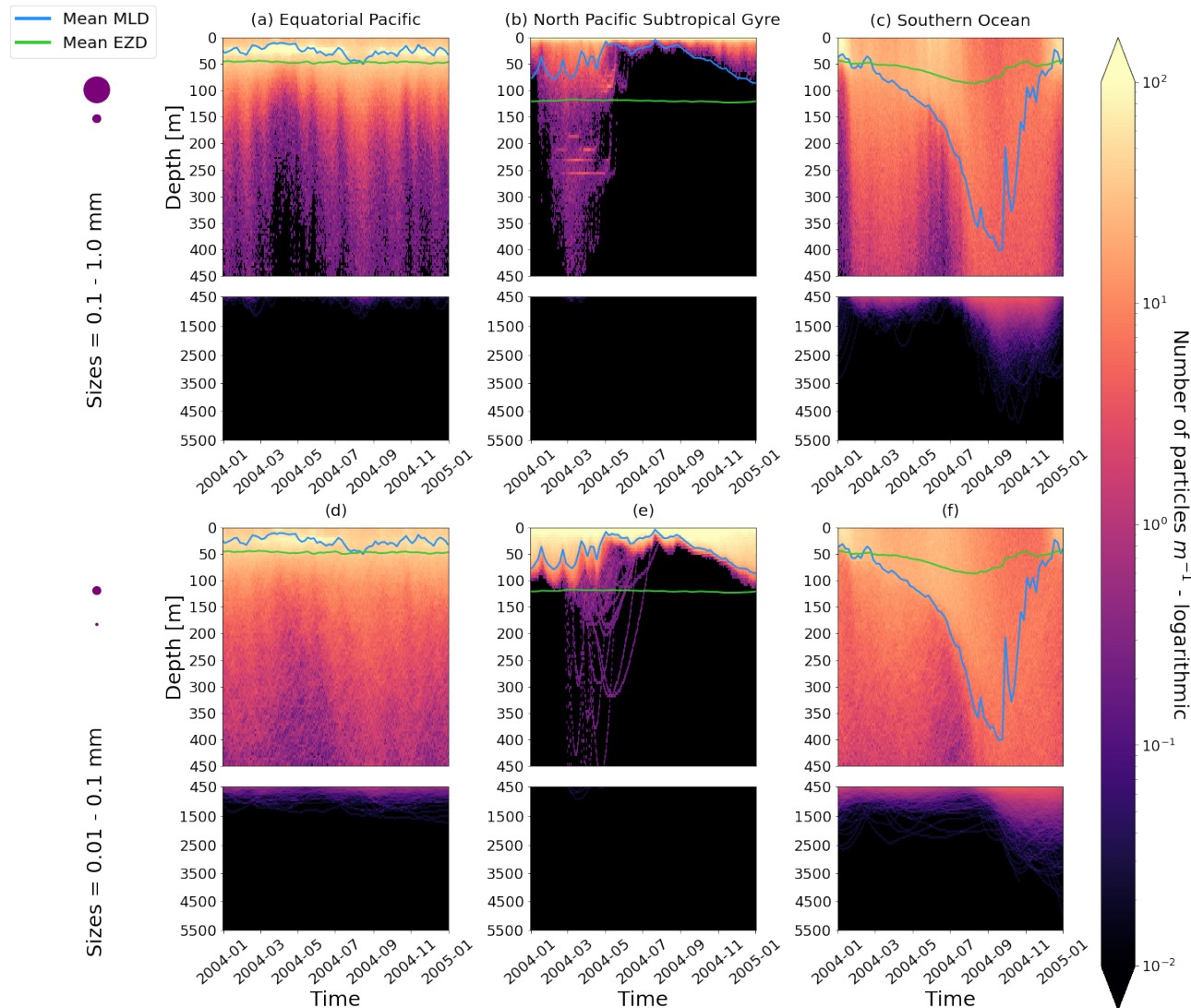

**Figure 2.** The vertical particle distribution of 10,000 particles over one year (after a three-month model spinup) for two size classes in three locations. The largest particles have a radius of 0.1 to 1.0 mm (top row; subplots (a), (b), (c)) and the smallest particles have a radius of 0.01 to 0.1 mm (bottom row; subplots (d), (e), (f)). The three locations are the Equatorial Pacific (EqPac; left column), North Pacific Subtropical Gyre (NPSG; middle column) and Southern Ocean (SO; right column). Each subplot is divided into two panels with the top panel from 0 to 450 m and the bottom panel from 450 to 5500 m. The horizontally averaged mixed layer depth (MLD; blue line) and euphotic zone depth (EZD; green line) are also displayed. Note that to isolate the effects of vertical physical characteristics and regional biological processes, horizontal advection is omitted from these simulations. The colorbar represents a logarithmic number of particles per meter (the total number of particles being 10,000), where light yellow denotes all values from 100 particles or more.

of 3D advection (Appendix E) and we demonstrate that even though particles can travel for thousands of kilometers (Fig. E1) after a few months (e.g. in the EqPac), the results are not largely impacted (Figs. E2–E4).

There are a few main results to highlight. Firstly, particles can sink far below the euphotic zone depth (EZD) and mixed layer depth (MLD) in regions with high biological activity (Fig. 2a and d; in the EqPac) and intense mixing (Fig. 2c and f; in the SO), and particles mostly stay above both the MLD and EZD otherwise (Fig. 2b and e; in the NPSG). Particles reach the deepest depths in the SO (with a maximum of >5000 m) and the shallowest depths in the NPSG (with a maximum of 1000 m). None of the particles reach the seafloor in the EqPac and NPSG, and only 15 of the 10,000 particles reach the ocean floor in the SO. Previous studies have found positively buoyant particles in sediment (Chiba et al., 2018; Peng et al., 2018) and Ye and Andrady (1991) had theorised that eventually particles can be fouled so heavily that they would permanently sink. We discuss potential reasons why so few of our particles reach the seafloor in Sect. 3.4. One of the key novelties of our study, introducing vertical wind-driven mixing, is to verify whether particles still oscillate as in Kooi et al. (2017) and Kreczak et al. (2021). We demonstrate that particles initiate their oscillations upon sinking below the mixed layer and in Sect. 3.3.1 we characterise these oscillations relative to particle size. In all three regions and for both size classes the average MLD seems to affect the vertical distribution of particles more than the average EZD. This is in contrast with the findings in Kreczak et al. (2021) that the EZD defines the vertical displacement of particles. Since they do not include advection or mixing in their model, this supports how important wind-driven mixing is for vertical displacement of particles <1 mm in the ocean. In general, in regions with more mixing (and less stratification), a particle that is moving vertically will be less affected by sudden changes in density and can sink deeper (for example, in the SO), whereas in regions with less mixing (more stratification), the opposite occurs and particles tend to sink to shallower depths (for example, in the NPSG).

### 3.1.1 High productivity region

In the EqPac, high biological activity means a biofilm can quickly form and increase the density of a particle. The most distinct feature is a subsurface maximum particle concentration throughout the year which generally appears just below the average MLD (around 30 m; Fig. 2a and d). The subsurface maximum is seen more prominently for the largest particles (0.1 to 1.0 mm) than the smallest ones (0.01 to 0.1 mm) and can reach a relative annual average concentration of 3 times the surface concentrations (Fig. 3a; up to 5 times considering the 95% percentile). Such an accumulation of particles at a certain depth can occur when the processes affecting the particles' upwards and downwards movement are in balance. In Kreczak et al. (2021), they hypothesise that in equatorial regions with large vertical density gradients, a 'subsurface plastic trapping layer' can be formed. Similarities can be drawn to the Choy et al. (2019) observational study; their samples were collected during months of high biological activity (January to April) and the Monterey Bay is an upwelling region; two features characterising our EqPac region. Their study also found a subsurface maximum concentration of particles below the MLD (albeit their average MLD was much deeper, at 200 m). The maximum depth reached by the smallest particles is around 1700 m, and the EqPac is the clearest example of larger particles oscillating with a higher frequency than the smaller ones. In a sensitivity analysis where we allow algal cell walls to remain attached to the particle after the biofilm dies, smaller particles no longer show a subsurface

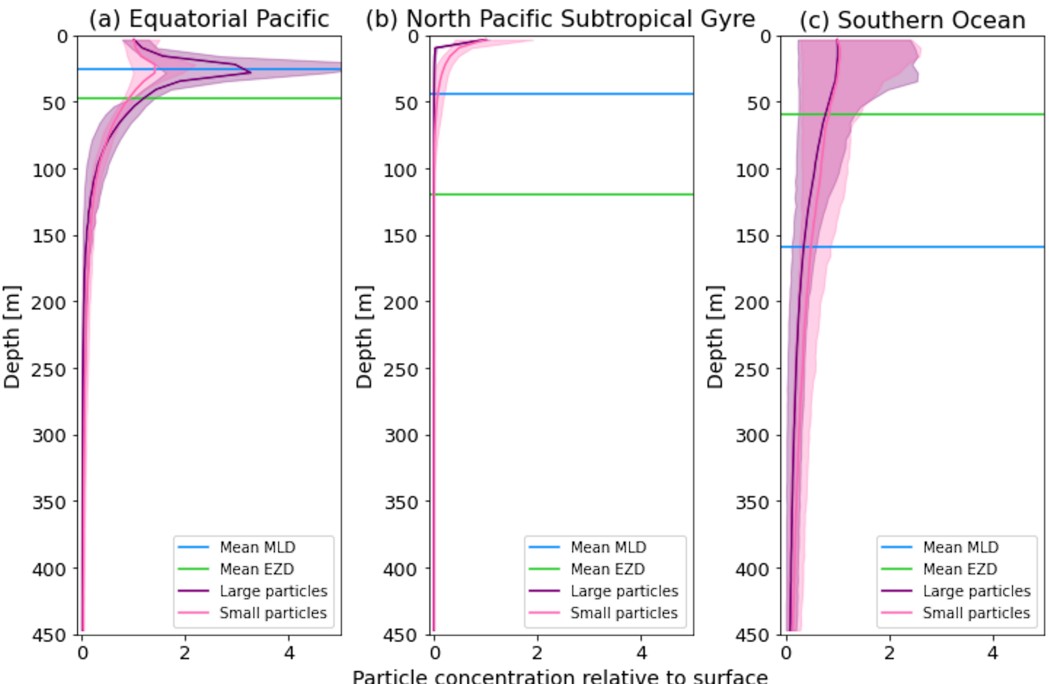

**Figure 3.** The annual average depth distribution of the trajectories in Fig. 2. Distributions down to 450 m for the three regions are shown: (a) Equatorial Pacific, (b) North Pacific Subtropical Gyre and (c) Southern Ocean. The x-axis shows the particle concentration relative to the surface at $z = 0.6$ m (where the surface concentration = 1, and 5 is 5 times the concentration of particles at the surface). Large particles (0.1–1 mm) are shown in purple and small particles (0.01–0.1 mm) are shown in pink. The 5-95% percentiles shaded around the mean are also shown, as well as the annual average mixed layer depth (MLD; blue horizontal line) and euphotic zone depth (EZD; green horizontal line) from Fig. 2.

maximum and instead can reach much deeper depths (to over 4800 m), since even a small change in density can affect its vertical transport and would cause it to sink deeper (Fig. D1d).

### 3.1.2   Low productivity region

In the oligotrophic NPSG, most of the particles of both size classes remain above the MLD throughout the year (Fig. 2b and e). Most of the largest particles remain at the sea surface, and the rest are distributed throughout the mixed layer (Fig. 2b).
This is clearly seen in the annual average distribution profile, where below 10 m the particles' concentration is almost 0 relative to the surface (Fig. 3b). The smallest particles in the NPSG are more evenly distributed from the surface to the base of the mixed layer than the larger particles (Fig. 2e; with an average annual MLD just above 50 m; Fig. 3b). These results are comparable to NPSG observations showing a power law decline in microplastic concentrations with depth, where most

particles are found in the upper tens of metres (Egger et al., 2020). Seasonality plays a role in the NPSG, where the boreal spring bloom (February to May) causes enough biofouling to occur for particles to sink below the MLD. A feature to highlight are a couple of horizontal lines representing subsurface maximum particle concentrations between 200 and 300 m in the larger particles (Fig. 2b). One possible explanation for this is an equilibrium between the biological and physical processes that can cause upwards or downwards movement of a particle. During a sensitivity test using an initial particle density of 30 kg m$^{-3}$ (representing expanded polystyrene) instead of 920 kg m$^{-3}$, the NPSG is the only region where vertical distribution changes drastically (Fig. C1b). Apart from during the spring bloom, all larger 30 kg m$^{-3}$ particles remain at the surface. Following results from Lobelle et al. (2021), this suggests that even under surface-mixing conditions, plastic with a radius of 0.1–1 mm and with a very low density might very rarely sink in oceanic regions with low algal concentrations. In the sensitivity analysis including the dead cell attachment as explained above, the smaller particles also have much longer oscillation times below the MLD, suggesting that the slower loss of biofilm mass affects the smaller particles much more than larger particles (Fig. D1e and b, respectively).

### 3.1.3    Intense mixing region

On average in the SO, particles mix to much deeper depths; the SO is the only region where the annual average particle depth distribution is not close to 0 at 450 m, relative to the surface (Fig 3c). The MLD varies greatly with the seasonal cycle (Fig 2c and f), gradually deepening from 50 m in the austral summer (January to March) to 400 m by the austral spring (October). The maximum depth reached by the particles in the SO (around 5000 m for both particle size classes) is in October for the largest size class. For the smallest particles, the maximum depth is delayed by a month or two due to the particles' longer oscillation lengths (further explored in Sect. 3.3.1).

Across all regions, the sensitivity analysis simulating the denser algal cell wall that remains attached after the biofilm is dead shows that oscillations still occur (Fig. D1). For the smaller size class, particles can reach deeper depths and longer oscillations, however the larger size class remains unchanged. Since this phenomenon has never been experimentally observed, here we suggest one alternative approach for the biofilm dynamics.

### 3.2    The influence of vertical advection and mixing

To visualise the processes that determine the vertical displacement of particles we display the ratio between the particles' absolute settling velocity, $w_s$ in Eq. (5), and the ambient water's vertical movement (Fig. 4). As described in the Methods section, the settling velocity is dependent on the initial size and density we assign to the particle and the biofouling dynamics, and the ambient water's movement includes vertical mixing ($w_a$; a combination of wind-driven and tidally-induced vertical mixing) and vertical advection ($w_a$). This means that particles present below the MLD are subject to vertical advection and vertical tidally-induced background mixing only (which are generally two orders of magnitude lower than wind-driven mixing in all regions; see Fig. A1–A3). Throughout all the simulations, two distinct horizontal layers are formed, one above the MLD, where the ambient vertical velocities dominate (and particles are strongly mixed, passively) and one below the MLD, where the particles' settling velocity dominates (and particles move actively, relative to the flow); Fig. 4. Below the MLD is where a

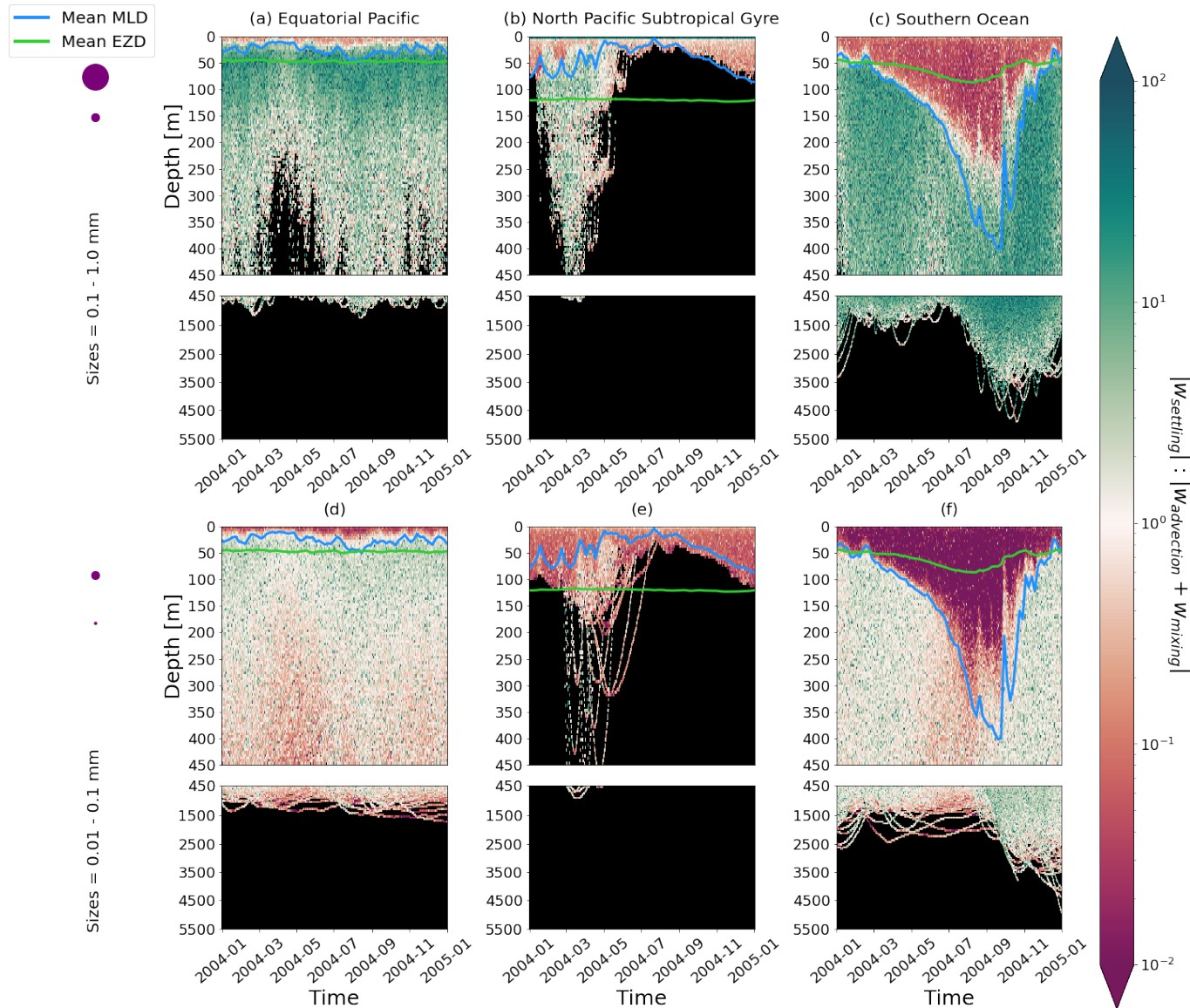

**Figure 4.** The ratio between a particle's absolute settling velocity ($w_{settling}$ or $w_s$ from Eq. (5)) and the absolute ambient vertical velocity; mixing plus advection; ($w_{advection} + w_{mixing}$ or $w_m$ from Eq. (3)) for each one-year trajectory from Fig. 2. Green patches represent instances where the particle's settling velocity dominates the vertical displacement of a particle, whereas red patches represent instances where physical ocean dynamics dominate. The same simulation duration, depth range, three regions, two size classes and MLD and EZD as in Fig. 2 are shown.

particle can oscillate due to sinking when the particle+biofilm's density exceeds surrounding seawater density and subsequent resurfacing once the biofilm's loss causes positive buoyancy to be restored (Kooi et al., 2017). At the maximum depth of the

oscillation, a short moment of neutral density leads to passive motion of particles (most visible for small particle trajectories; in red in Fig. 4d, e, f). Since larger particles oscillate more than smaller particles, the ratio between the absolute settling velocity and the ambient vertical velocities is larger (the green patches in Fig. 4a, b and c compared to d, e and f). Since smaller particles are more easily mixed above the MLD, the ratio of ambient vertical velocities to the settling velocity is larger (the red patches in Fig. 4d, e and f compared to a, b and c). Areas where the ratio is one show depths at which both the ambient velocity and the relative velocity are important to determine the vertical motion of the particles.

## 3.3 The dominant depth-dependent processes in the biofouling dynamics

We also analyse the results by identifying which of the five biofilm gain and loss terms from Eq. (6) dominate for different depths, particle sizes and regions of our study (Fig. 5). Biofilm growth ($G_{grow}$) mostly dominates from the surface down to the base of the mean EZD (around 50 m) in the EqPac and SO (Fig. 5a, c, d, f). Below that, down to about 100 m, grazing ($L_{graze}$) becomes the largest term, apart from when the MLD is very shallow (March to May in EqPac; Fig 5a and d) or very deep (April to December in SO; Fig 5c and f). Below 100 m in the EqPac and SO, loss of the biofilm via respiration ($L_{resp}$) is dominant (resulting in oscillations, characterised in the next section). Nonlinear losses ($L_{nonlin}$) only dominate for the month of December in the SO between 50–100 m potentially because there could be slightly less grazing during the peak of the austral summer months.

The NPSG particles show rather different dominant patterns than the other two regions. Throughout most of the year, particles do not reach the very deep euphotic zone (>100 m), hence $G_{grow}$ is the dominant term down to the MLD. Due to the smallest particles being most affected by the modelled wind-driven mixing, collisions ($G_{coll}$) end up dominating the biofilm dynamics around the base of the MLD from June to March (Fig. 5e). The same feature is slightly visible for larger particles, although not as prominently (Fig 5b). During the spring bloom, $G_{grow}$ dominates down to the base of the EZD, as well as between around 150 and 250 m for the larger particles (Fig. 5b). To explain this, we looked at the profiles of dissolved inorganic nitrate from NEMO-MEDUSA (not shown), where the surface is nutrient-depleted but below 100 m, concentrations increase linearly. This means that between 150 and 250 m there is sufficient nitrate for the biofilm to grow, but light is limiting and at the surface the opposite is true (nutrients are limiting but there is sufficient light). This results in $G_{grow}$ having a similar order of magnitude at the surface and 150-250 m.

We can also sum the mass flux of each of the five terms over the entire trajectories for each region, to compare the overall dominance of each term (Fig 6). Since most of the particles stay within the top 50 m in all three regions, and $G_{growth}$ dominates at these depths (as seen in Fig 5), 50% of the one-year biofilm mass flux is controlled by growth. The next largest overall term is $L_{resp}$, for 30% of the data points in the EqPac and SO, and almost 50% in the NPSG. Finally the $G_{coll}$, $L_{graze}$ and $L_{nonlin}$ are all below 15% in all three regions (with collisions almost at 0).

### 3.3.1 Characterising oscillations

As in Kooi et al. (2017), particles in our model oscillate in the water column. As described above, as soon as a particle sinks below the euphotic zone, algal loss via grazing and respiration dominate the biofilm dynamics. This results in the particle+biofilm

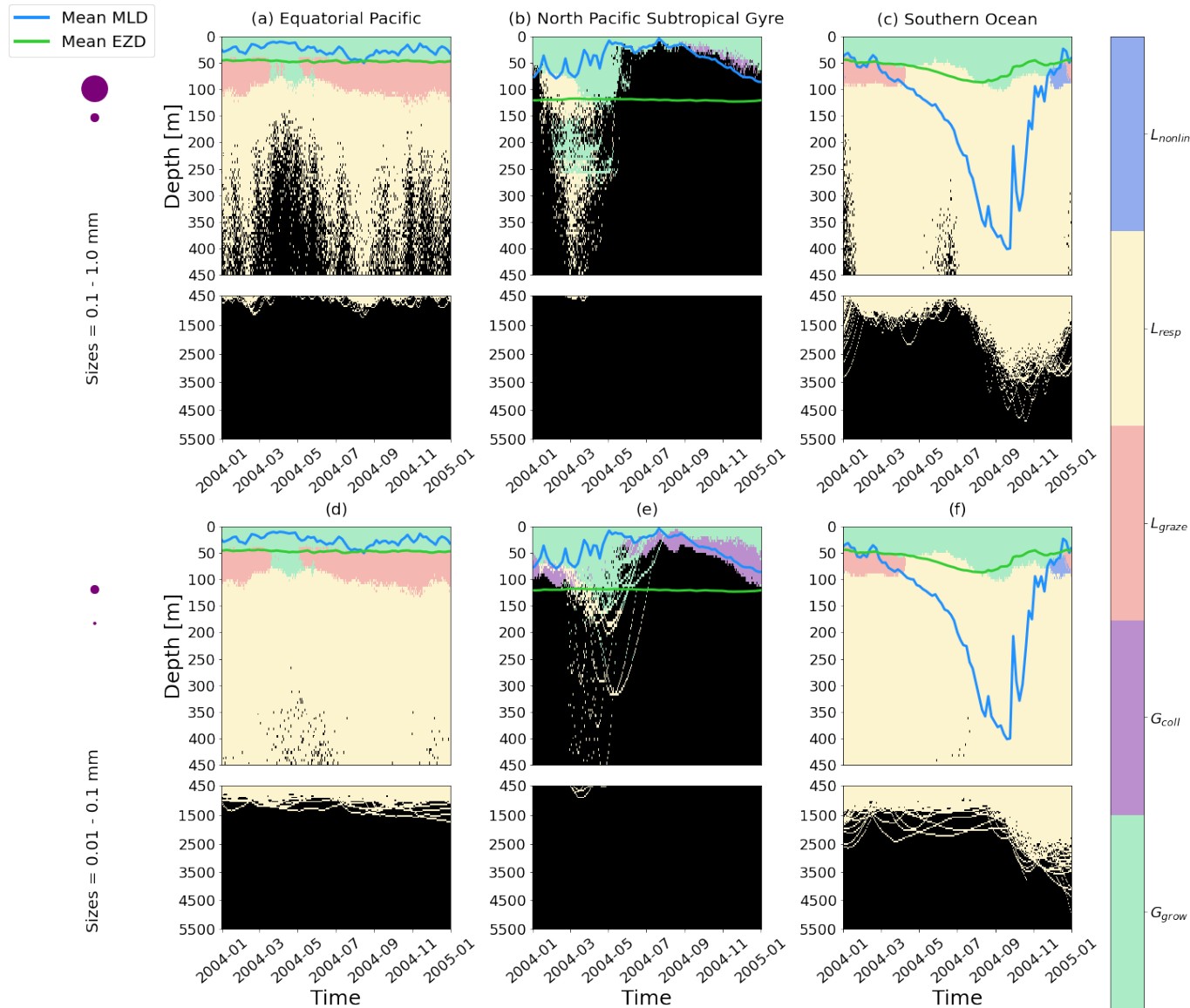

**Figure 5.** The dominant term out of the five terms that determine the gains and losses of biofilm dynamics on the plastic particles from Eq. (6), over the one-year trajectories in Fig. 2. The three loss terms are nonlinear loss ($L_{nonlin}$) in blue, respiration ($L_{resp}$) in yellow and grazing from mesozooplankton ($L_{graze}$) in orange. The two biofilm gain terms are via collisions with planktonic diatoms ($G_{coll}$) in purple and growth ($G_{grow}$; primary productivity) in green. The horizontally averaged mixed layer depth (blue line) and euphotic zone depth (green line) are also displayed. The same simulation duration, depth range, three regions and two size classes as in Fig. 2 are shown.

eventually reaching a depth where it no longer has a density exceeding that of the surrounding seawater and rising. A point that Kooi et al. (2017) makes regarding different sizes of particles is also supported in our results; the smaller the particle, the

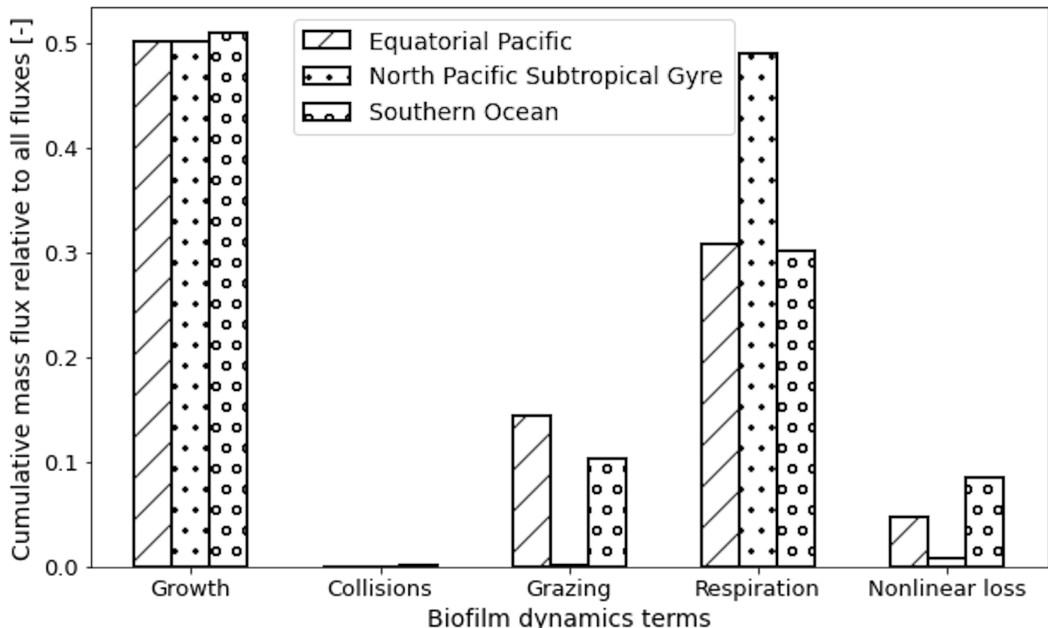

**Figure 6.** The overall relative cumulative mass flux of each of the five terms from Eq. (6) and Fig. 5 using all trajectories in the three regions of our study; (a) Equatorial Pacific (forward slashes), (b) North Pacific Subtropical Gyre (dots), (c) Southern Ocean (circles).

lower the frequency of its oscillation. Larger particles have a higher sinking velocity than smaller particles (see green bars in
Figs. A1–A3 and Fig. 3 in Kooi et al., 2017) and hence sink and rise faster than smaller particles. This is represented clearly
in the EqPac and SO (Fig. 7a and c), where oscillation lengths can reach about 220 days (99th percentile) in the EqPac for
particles with a radius of 0.01 mm and about 280 days (99th percentile) in the SO. In the NPSG, since very few particles
sink below the mean EZD and therefore do not oscillate, this pattern of smaller particles having longer oscillation lengths
than larger particles is not seen (Fig. 7b). The oscillation length for NPSG particles is just under 100 days (99th percentile),
probably occurring during the 3 months of the spring bloom when particles do sink below the EZD (Fig. 5b and e). In all three
regions, for particles of 1 mm, the mean oscillation length is less than 10 days. One of the key differences to the Kooi et al.
(2017) results is that the oscillations reach much deeper depths. The main reason for this is that the Kooi et al. (2017) study
uses a very shallow EZD (around 20 m) and with their limitation of using a constant grazing term at all depths, the biofilm
dies and resurfaces very rapidly below the EZD. It should also be noted that since respiration is the dominant process below
the MLD in general, and respiration is dependent on temperature (Eq. 8), the oscillatory behaviour of particles is dependent on
the surrounding water temperature (Appendix F). The respiration rate decreases exponentially with a decrease in temperature,
so therefore a biofilm in deeper, colder water has lower respiration rates than the shallower particles in (Kooi et al., 2017).

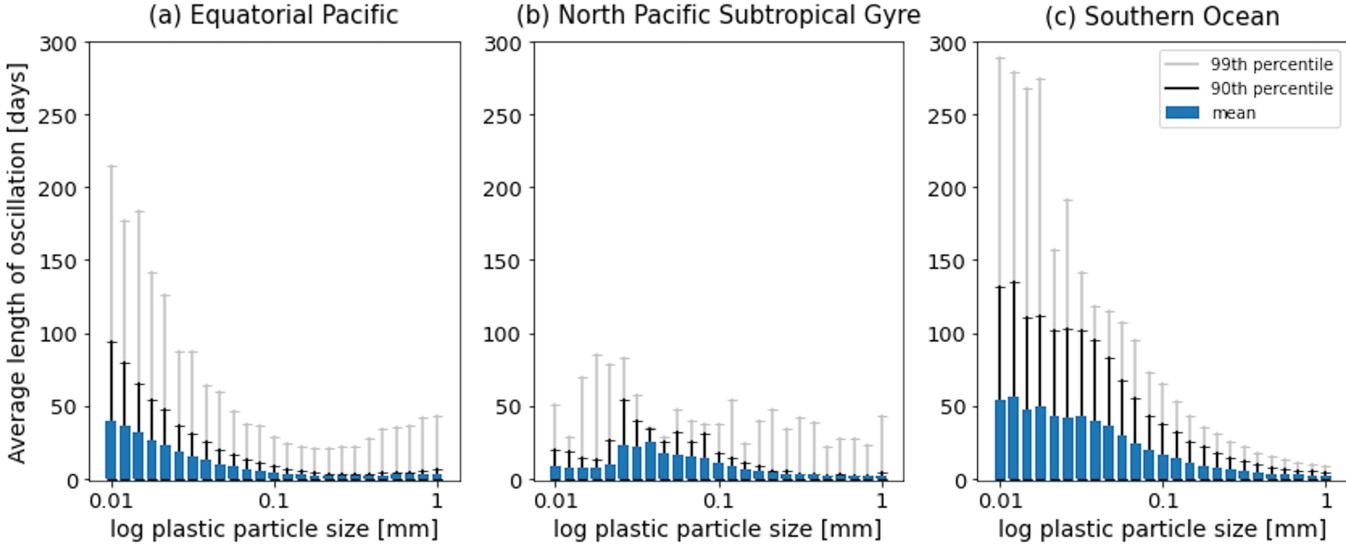

**Figure 7.** The average length [days] of the oscillations over the one-year simulations in Fig. 2. Results for the 25 size bins (particles with a radius between 0.01 and 1 mm) for the three regions are shown: (a) Equatorial Pacific, (b) North Pacific Subtropical Gyre, (c) Southern Ocean. The 90th percentile for each size bin is represented by the black line and the 99th percentile is represented by the grey line.

### 3.4 Model assumptions and future model developments

As with all models, our results depend on our parametrisations, assumptions and model design. Firstly, as explained in the previous section, the model relies strongly on the assumption that biofilm respiration depends only on temperature. After a particle is biofouled and sinks, continued respiration is the main mechanism for defouling, which in turn leads to the oscillation of the microplastic. Such behaviour is still theoretical and has never been experimentally observed. Furthermore, any biofilm on the particle is assumed to be denser than water density, though this has only been observed in coastal waters (Amaral-Zettler et al., 2021b).

The other main assumption is that our modelled biofilm only consists of photosynthetic algae (diatoms), while observed biofilm community structures are shown to vary both spatially (e.g. between the Atlantic and Pacific; Amaral-Zettler et al., 2015) and temporally (Amaral-Zettler et al., 2020). In the latter study, diatoms dominate within the first week of colonisation of microplastic, after which other groups, including Rhodobacteriaceae, become more prevalent. Other observations show that diatoms are still one of the most frequently found species after 14 weeks, though the total biofilm species richness is high (Bravo et al., 2011). Furthermore, Amaral-Zettler et al. (2021b) mention that the plastisphere consists of animals and heterotrophic protists that would not necessarily be impacted by the dark and as diatoms die, this could actually stimulate the growth of heterotrophs, affecting the particle's buoyancy in different ways. All these studies show that community and food web dynamics are complex, chaotic and thus fundamentally indeterminate. Therefore, even if a biogeochemical general circulation model had more algal and bacterial species available, it would be very challenging to model the exact community

composition of a biofilm at a particular point in time and space. Nevertheless, one possibility for future work is to apply a distribution function to biofilm density, growth and death parameters and use a Monte Carlo sampling approach to model these parameters probabilistically. Though this could be computationally demanding, a sensitivity analysis for a simplified scenario could be the starting point. Kreczak et al. (2021) perform a sensitivity analysis using the quotient between algal growth and death rates, for example, and this type of study could be expanded to include the full community diversity of a biofilm.

Some of the biofouling dynamics could also be further developed in future work. For example, we currently assume that the attached algal mesozooplankton grazing rate and growth rate are the same as planktonic algae. We have made this assumption because little is known about the dynamics of plastic biofilms from laboratory settings or in situ observations. Our sensitivity test regarding algal frustules remaining attached to the particle once the biofilm dies (Appendix D) could be improved by using data from biofouling experiments in the dark. One could even test the effects of diatoms entering a dormant phase without nutrients or light. Another aspect to investigate is whether the colonisation or growing of cells could result in a boundary layer effect in which nutrient supply is reduced for biofilm cells below the newly attached cells. Further, the fact that the nonlinear diatom losses in NEMO-MEDUSA include grazing by unmodeled higher trophic levels suggests that the entire plastic plus biofilm could be ingested (instead of individual algal cells within the biofilm, as we assume here). This could be addressed by coupling our biofouling model with other modeled biological interactions such as ingestion (e.g. Cole et al., 2013; Kvale et al., 2021) and egestion of plastic or the merging of particles to model plastic trapped in marine snow. Again, a probabilistic analysis, as described above, could be one approach. Lastly, further sensitivity analyses can be carried out regarding the collision rate, growth rate and other parameters that the model performance relies on, since analysing the full combination of all ranges of parameters was beyond the scope of this study.

We only use spherical plastic particles since the equations to determine their settling velocity only apply to spheres (Kooi et al., 2017) and a universal model that fits all (micro)plastic types is currently unfeasible to derive (Kreczak et al., 2021). A recent study shows that fibers make up a significant part of plastic particles in aquatic environments and their high surface area to volume ratio leads to a larger area for contact with biology (Kooi et al., 2021). Biofouled ellipsoid-shaped particles could become denser faster than spheres (which have the smallest surface area for any given volume), meaning they could possibly reach the seafloor before the biofilm dies. We hypothesise that upon including different shapes and biological interactions (described above) we might see some vertical distributions such as in Peng et al. (2018), where concentrations of plastic <5 mm are several times higher in the deep-sea (Mariana Trench) than near-surface waters. The Kooi et al. (2017) model also places limits on the maximum size of the particles, as de la Fuente et al. (2020) show that for particles of size 1 mm, the Maxey-Riley-model is only valid for sinking velocities below 2 cm s$^{-1}$; a condition that is just about met below the mixed layer in our simulations (green bars in Figs A1–A3).

Being a process study, we have chosen not to include horizontal advection in simulations so that the particles do not move away from regions with defined biophysical profiles that we are interested in (though we did add a sensitivity analysis in Sect. E). Future work with different aims, such as estimating where particles (that are subject to biofouling) end up when releasing them from source locations, would benefit from having 3D advection incorporated.

Finally, as mentioned in the Introduction, there have been very few subsurface plastic concentration observations to date. Overcoming some of the logistical challenges to measure and monitor the 3D movement of plastic (specifically smaller than 1 mm) in the open ocean is becoming urgent in order to validate that: (1) biofouled particles oscillate (since this has never been observed) and (2) their distribution is similar to our modeled results (down to 5000 m) in regions with similar biological and physical properties.

## 4  Conclusion

We have explored the vertical distribution of ocean plastic spherical particles between 0.01 and 1 mm that are initially buoyant and have been submerged due to biofouling. We present one-year trajectories with more realistic physical and biological dynamics than in the Kooi et al. (2017) and Lobelle et al. (2021) models. The three regions in this study (Equatorial Pacific, North Pacific Subtropical Gyre and Southern Ocean) represent areas in the ocean with different biological activity and wind-induced mixing which have varying impacts on the average vertical distribution of particles. In upwelling regions with high productivity and algal concentrations (the EqPac in our study), a subsurface maximum particle concentration is present just below the climatological MLD (which can reach up to five times the concentration of the surface). In regions with very low productivity and low wind-induced mixing (the NPSG in our study), particles remain in the upper few tens of metres and can only sink below the mixed layer if a spring bloom occurs. In areas with very high wind-induced mixing and hence a deep mixed layer (down to 400 m in the Southern Ocean in our case), particles can reach depths of thousands of metres (about 5000 m).

This model has been developed in order to gain further understanding of the mechanisms driving the vertical distribution of marine plastic particles. Our model can be incorporated into more sophisticated model studies mapping the total global budget of marine (micro)plastic debris. It can therefore be coupled to models that wish to, for example, add other species to the biofilm community, add sinking due to marine snow and fecal pellet aggregates, as well as how biofouling can affect the weathering of particles and sorption or release of persistent organic pollutants (Rummel et al., 2017). Finally, our model can provide estimations for subsurface concentrations if surface samples have been collected and the full-depth biophysical properties are known.

*Code availability.* The full code to run our simulations is accessible from: https://github.com/OceanParcels/biofouling_3dtransport_2.

**Appendix A:  Isolating the three vertical velocities present in the model per region**

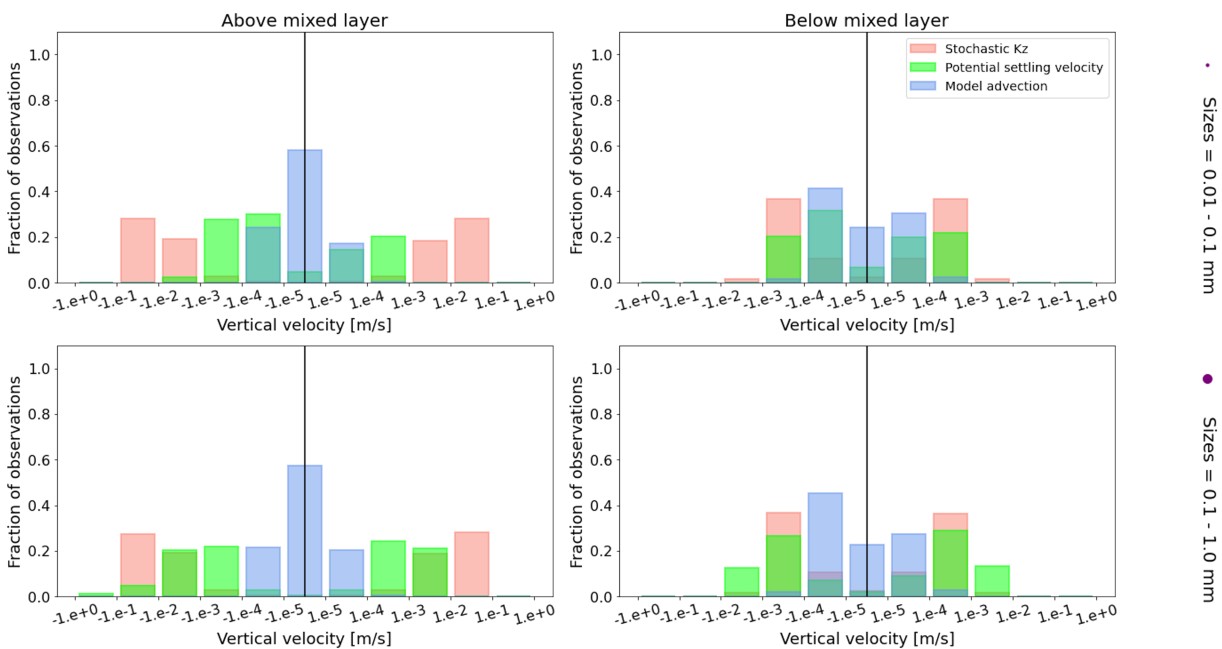

**Figure A1.** Equatorial Pacific's annually averaged vertical velocities [m s$^{-1}$]; stochastic Kz (red) which comprises of tidally-induced mixing and wind-driven mixing, potential settling velocity of the particle (green) which is $w_s$ in Eq. (5) and NEMO-MEDUSA vertical advection (blue). The top row is for particles of the smallest size class (0.01–0.1 mm) and the bottom row is the largest size class (0.1–1 mm). The left column is for particles within the wind-mixing region (above the mixed layer) and the right column is for particles below the mixed layer. The vertical black line represents a vertical velocity of 0; to the right of that line is for upward velocities and to the left is for downward velocities, each bar indicating an increase by an order of magnitude.

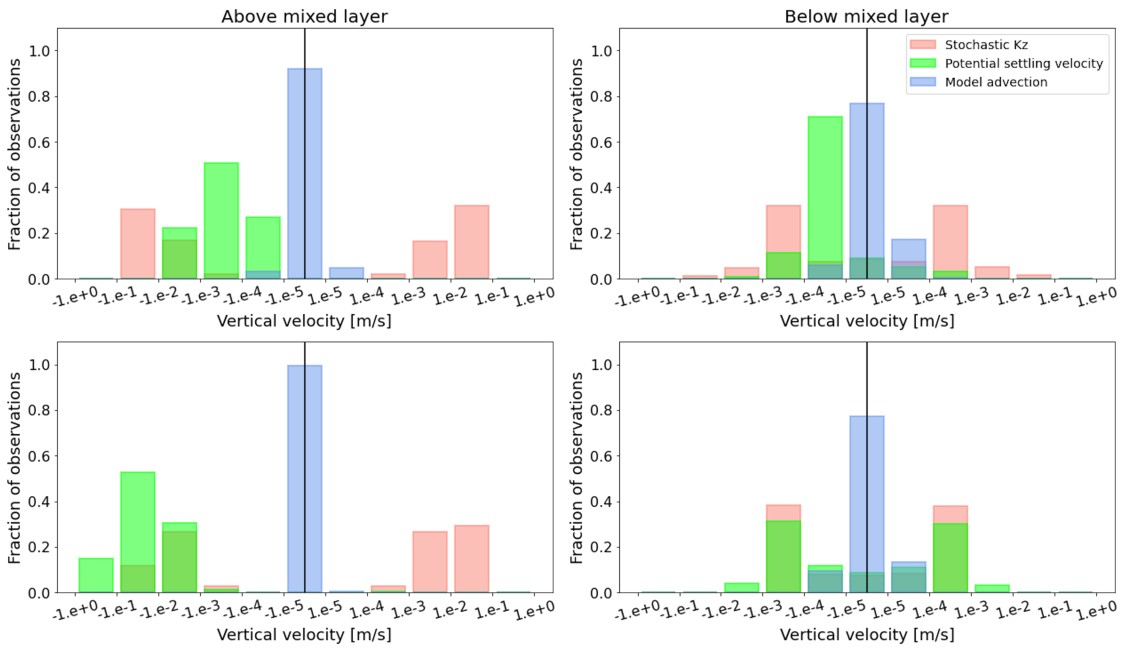

**Figure A2.** As in Fig A1 but for the North Pacific Subtropical Gyre.

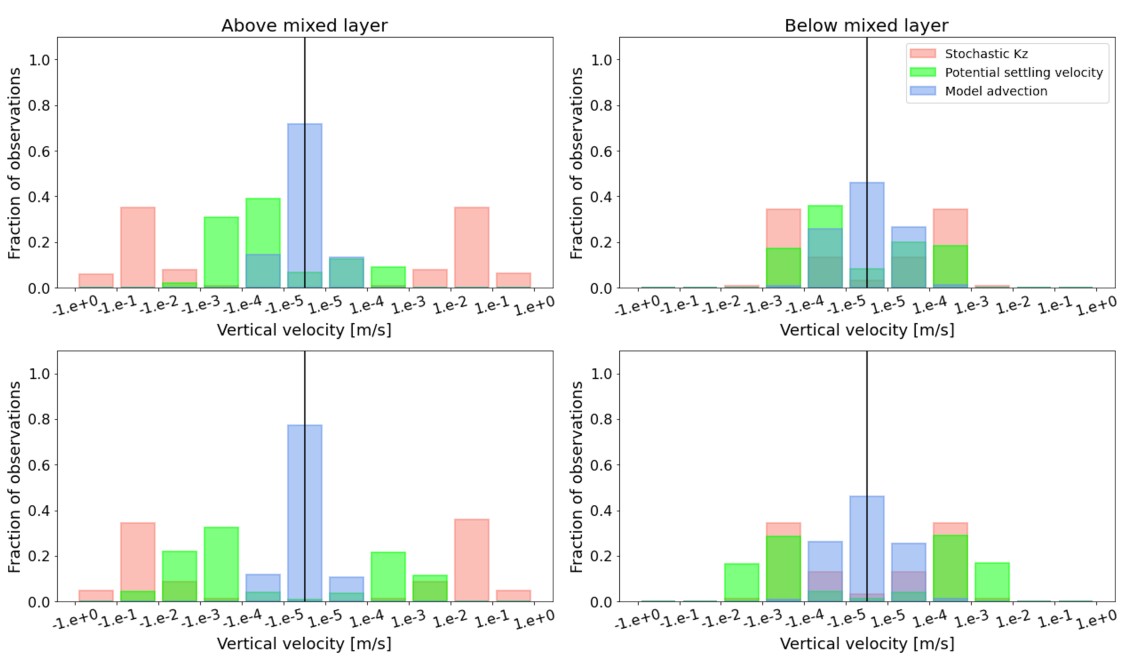

**Figure A3.** As in Fig A1 but for the Southern Ocean.

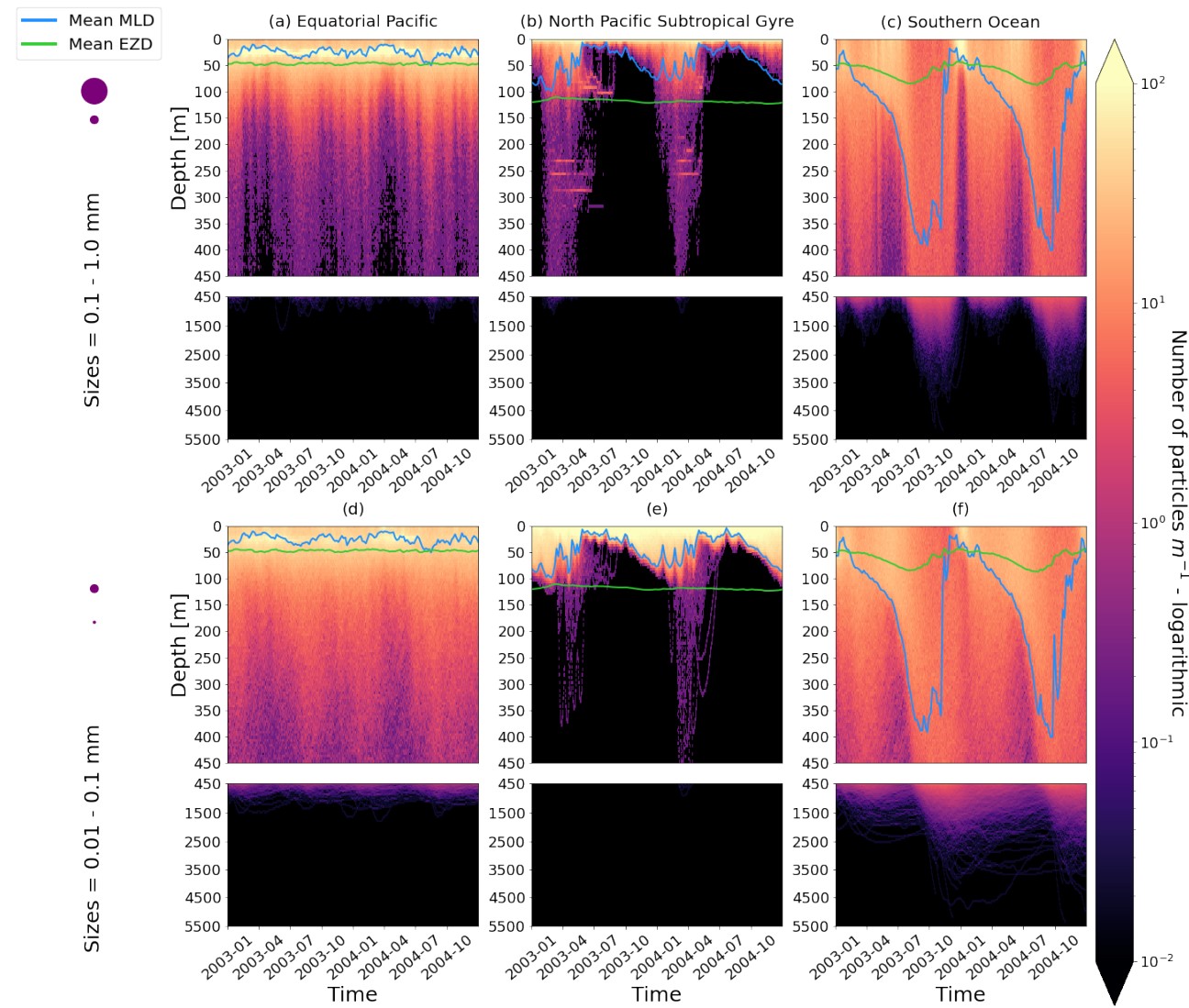

**Figure B1.** As in Fig 2 but for 2 years (from 1 January 2002 to 31 December 2004).

## Appendix B: Two-year simulations

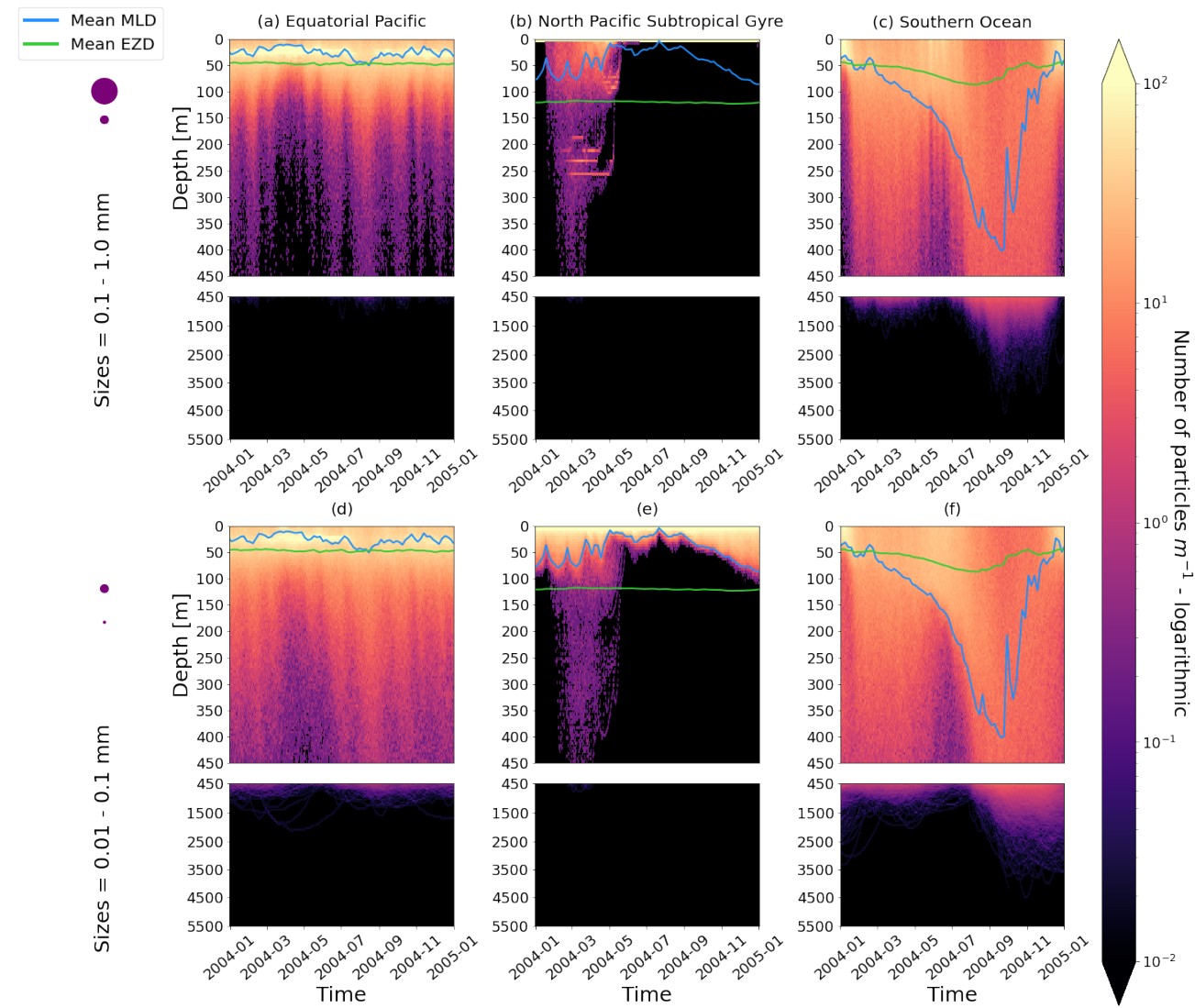

**Figure C1.** As in Fig 2 but for particles with an initial density of 30 kg m$^{-3}$, representing expandable polystyrene.

**Appendix C: The effect of changing the initial density of the particle**

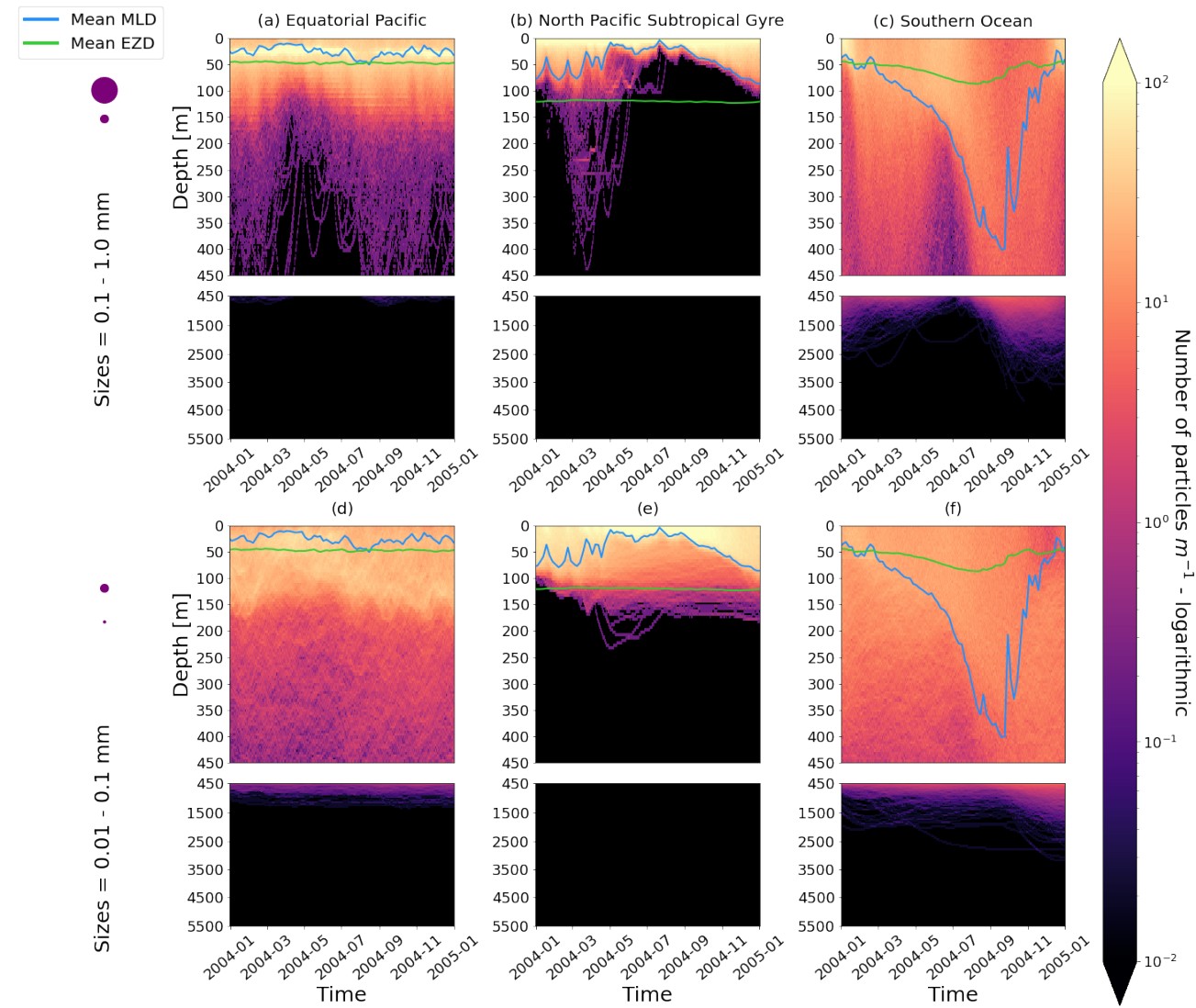

**Figure C2.** As in Fig 2 but for particles with an initial density of 1020 kg m$^{-3}$, representing rigid polyamide.

## Appendix D: Modeling how dead biofilm cells remain attached in the dark

We have also explored alternative assumptions for the biofilm dynamics. Specifically, we want to see what happens to trajectories once we allow for some of the biofilm to remain attached in the dark (below the EZD). Algal cells are composed of an organic interior (the cytoplasm) and an opaline silicon cell wall (the frustule). We hypothesise that the frustule could remain attached in the dark. Miklasz and Denny (2010) report that the frustule can be denser than the cytoplasm (with a frustule's median density of 1800 g m$^{-3}$ vs 1065 g m$^{-3}$ for the cytoplasm). This alternative model means that a particle could keep sinking once the biofilm is dead due to the attached frustules. We simulate that the living algal loss terms ($L_{graze}$ in Eq. (7), $L_{resp}$ in Eq. (8) and $L_{nonlin}$ in Eq. (9)) is proportional to the gain of mass of the frustule, hence:

$$\frac{\mathrm{d}A_{fr}}{\mathrm{d}t} = L_{graze} + L_{resp} + L_{nonlin} - L_{diss}, \tag{D1}$$

where $L_{diss}$ is the dissolution of algal cell walls (0.006 d$^{-1}$; Yool et al., 2013). Another important component of this model is that the median radius of the frustule in Miklasz and Denny (2010) is $\frac{1}{60}$th of the radius of the cytoplasm. We include this into the model by adapting Eq. (8) from the Kooi et al. (2017) model as follows:

$$v_{cy} = \frac{4}{3}\pi \left( r_A * \frac{59}{60} \right)^3, \tag{D2}$$

where $v_{cy}$ is the volume of the cytoplasm [$m^3$] and $r_A$ is the radius of an algal cell [m]. The volume of the frustule ($v_{fr}$) is therefore $v_A$-$v_{cy}$ and the volume of the whole dead biofilm ($v_{bfdead}$) is $v_{fr}*A_{fr}*\theta_{pl}$, where $\theta_{pl}$ is the surface area of the plastic particle [$m^2$]. The total volume of the plastic particle plus the biofilm (Eq. (7) in Kooi et al., 2017) is then: $v_{tot} = v_{pl} + v_{bfdead} + v_{bfA}$, where $v_{pl}$ is the volume of the plastic particle [$m^3$] and $v_{bfA}$ is the volume of the living biofilm ($v_{bfA} = (v_A * A) * \theta_{pl}$). The equations following these adaptations are then as described in the supplementary information in Lobelle et al. (2021).

As mentioned in the main text, the effect of this alternative model on larger size classes is almost negligible, probably due to this last component, that the radius of the frustule is so much smaller than that of the cytoplasm. The smallest size class, however, is affected due to the fact that the frustules could reach the sizes of the plastic particles and hence affect the density difference between the particle plus frustule and surrounding seawater. It is important to note, however, that since little is known about what happens to biofilms in the dark, we have used some basic assumptions in constructing this model addition and these results should be considered with caution.

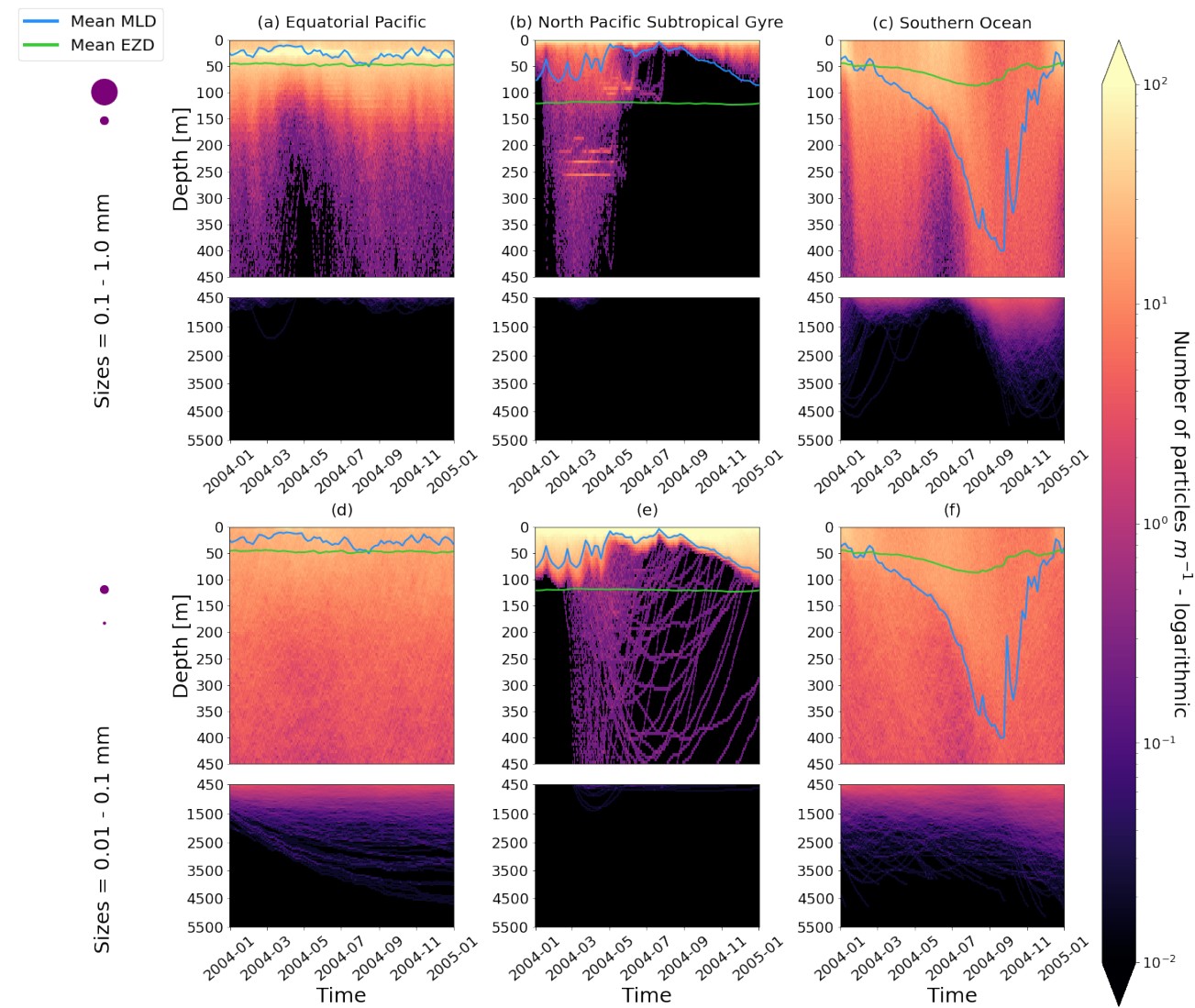

**Figure D1.** As in Fig. 2 but for particles where the biofilm death allows for denser algal frustules to remain attached (see description in Appendix D). The trajectories mostly remain unchanged for the larger size-class, however the smaller size-class has deeper trajectories and longer oscillation timescales.

**Appendix E: The effect of including 3D advection**

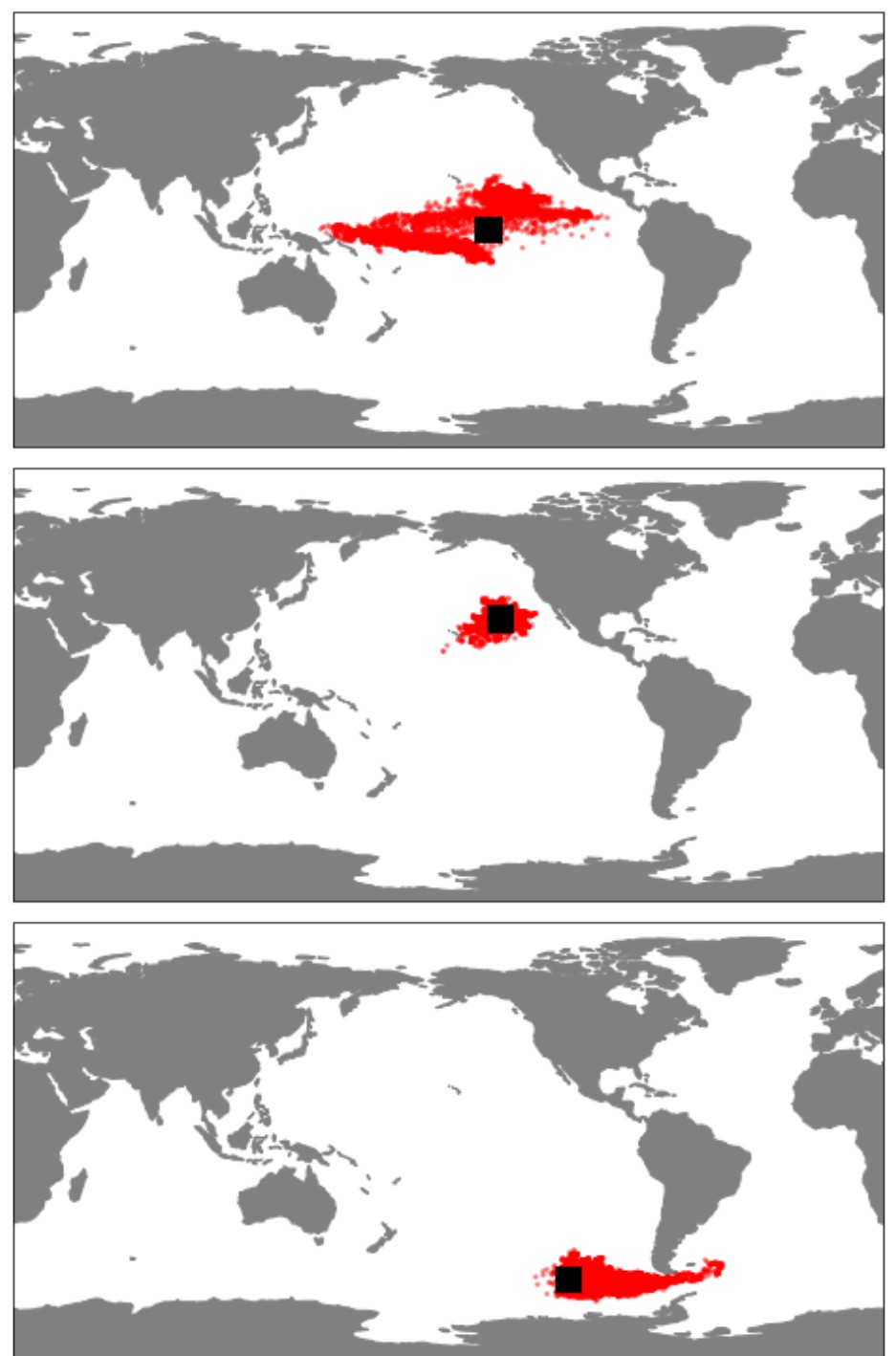

**Figure E1.** The horizontal trajectories of particles released in the three regions of our study (a) EqPac, (b) NPSG and (c) SO when 3D advection is included. The simulations ran for 180 days and the black squares represent the initial release location.

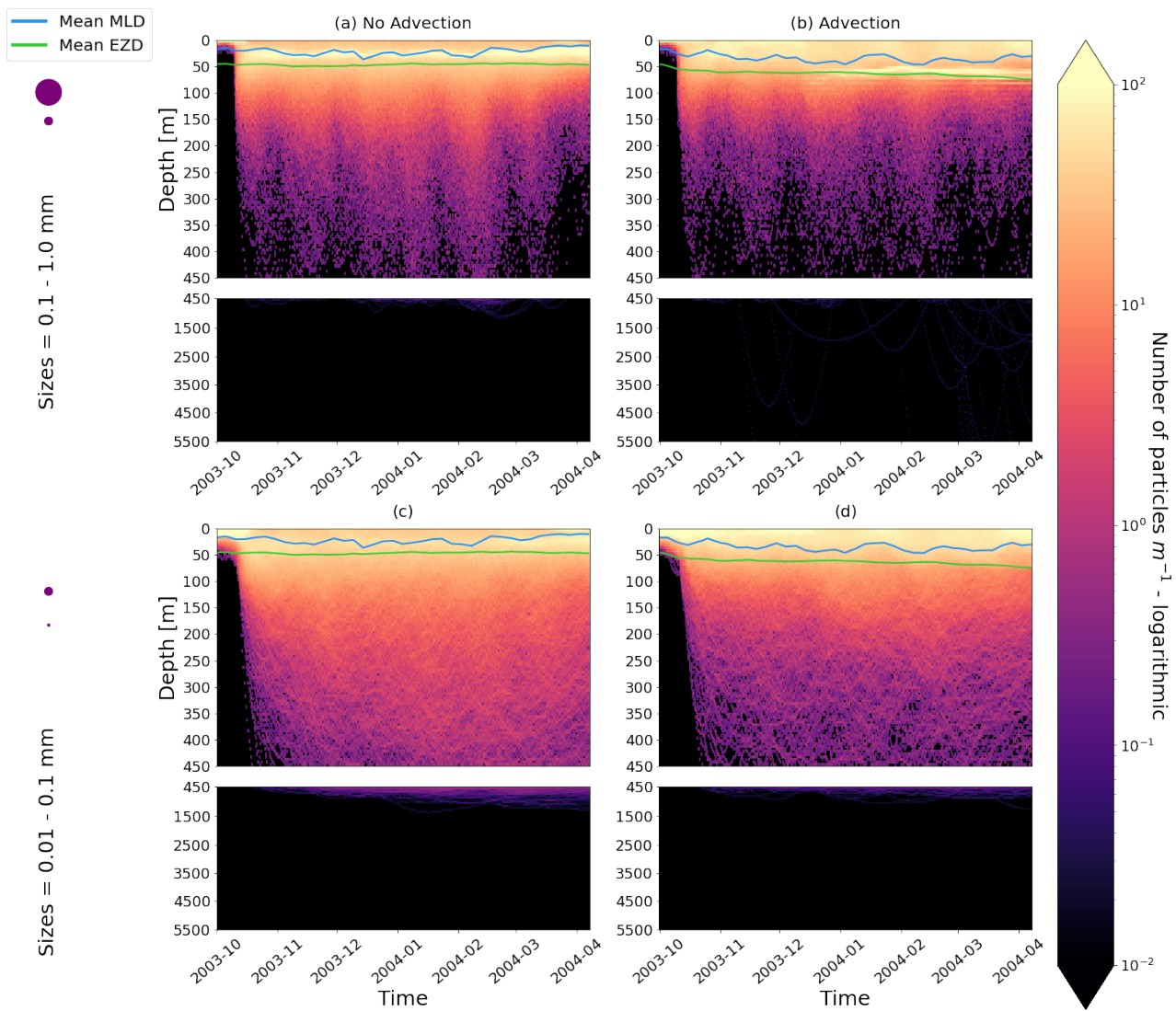

**Figure E2.** As in Fig 2 but for particles only in the EqPac and running for 180 days. The left 2 figures are without advection and the right 2 figures are with advection. This means that (a) and (c) are identical to Fig 2a and Fig 2d, however, including the 3-month spin-up time and running for fewer days (not a full year).

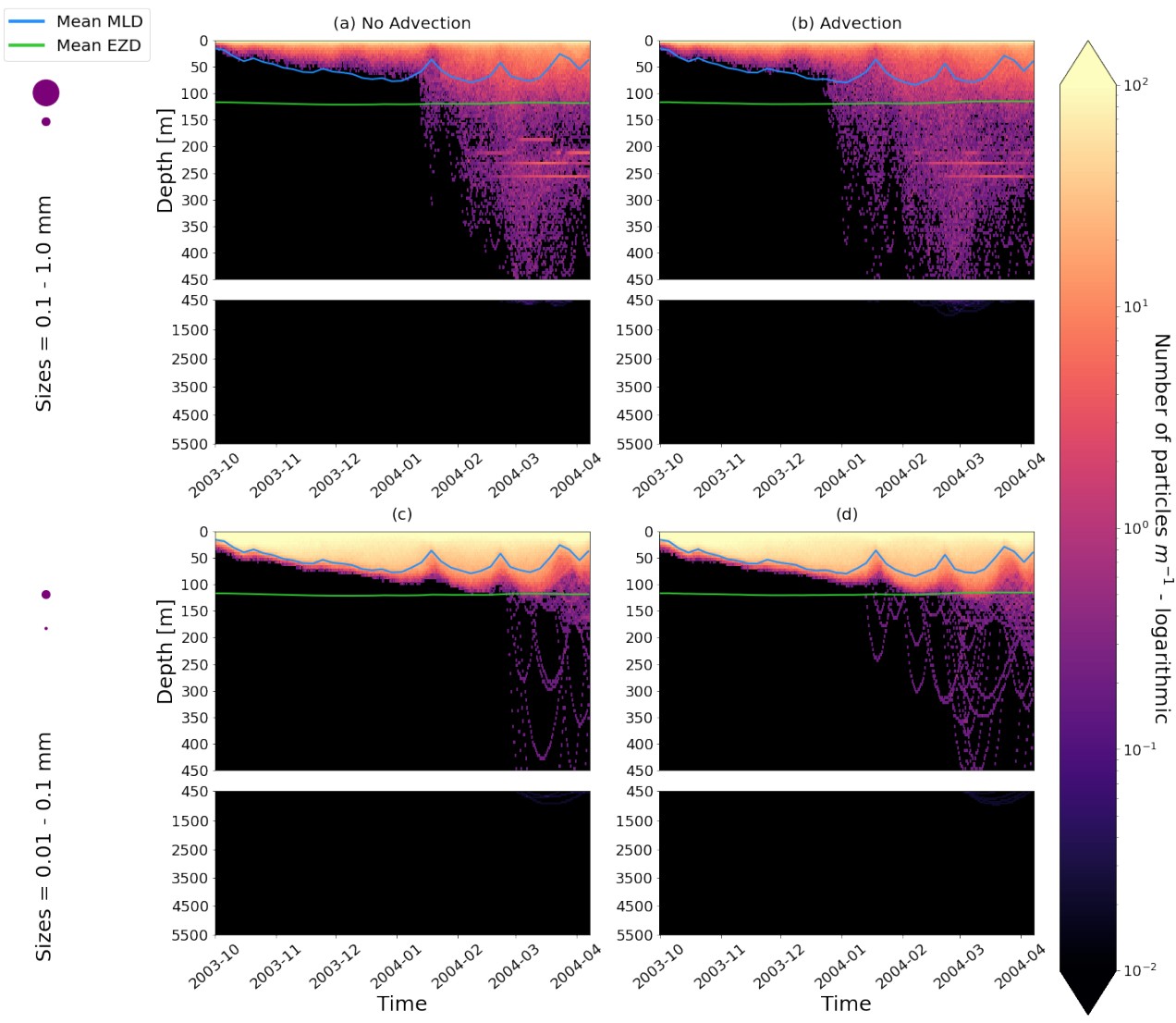

**Figure E3.** As in Fig E2 but for particles only in the NPSG. This means that (a) and (c) are identical to Fig 2b and Fig 2e, however, including the 3-month spin-up time and running for fewer days (not a full year).

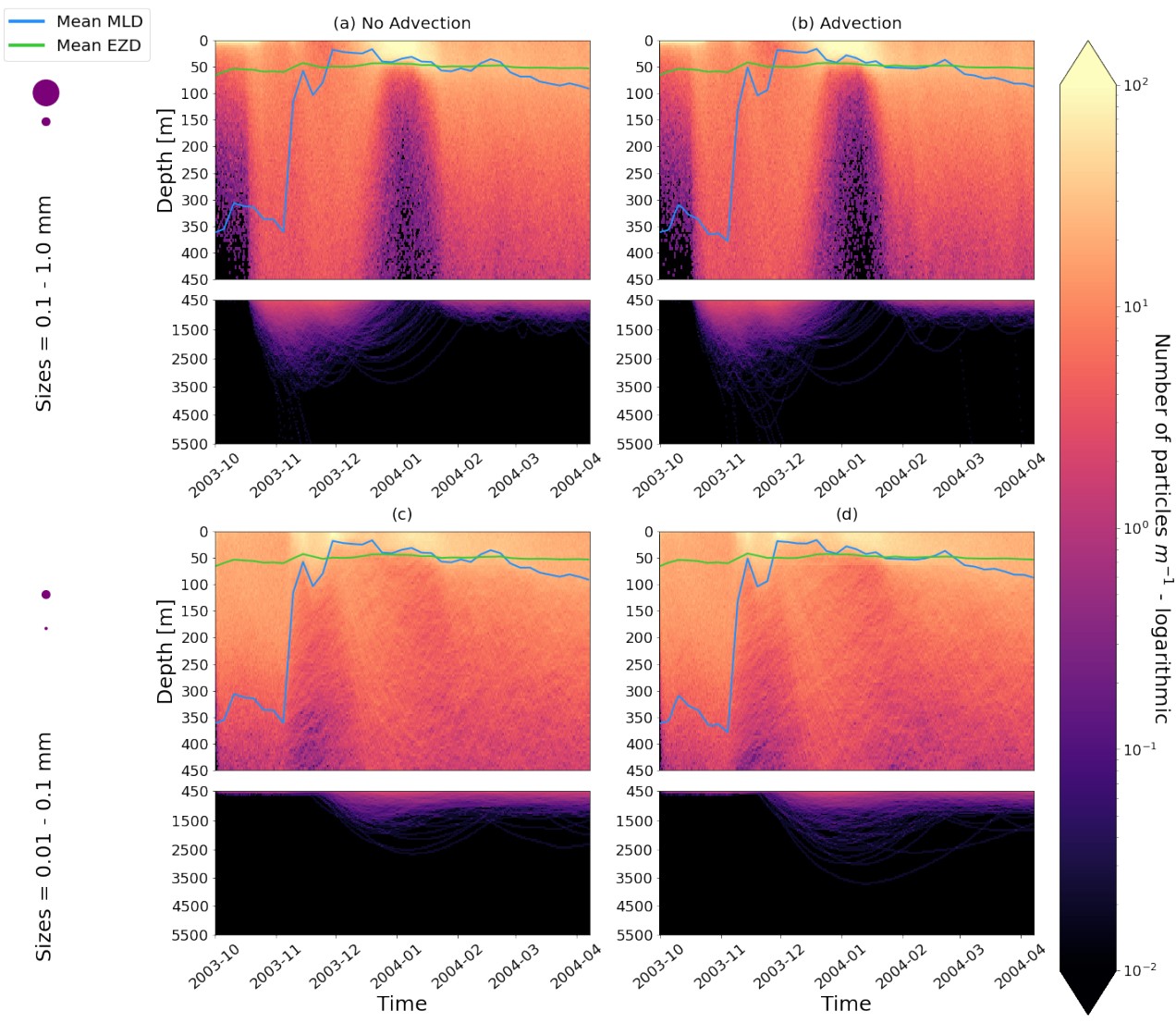

**Figure E4.** As in Fig E2 but for particles only in the SO. This means that (a) and (c) are identical to Fig 2c and Fig 2f, however, including the 3-month spin-up time and running for fewer days (not a full year).

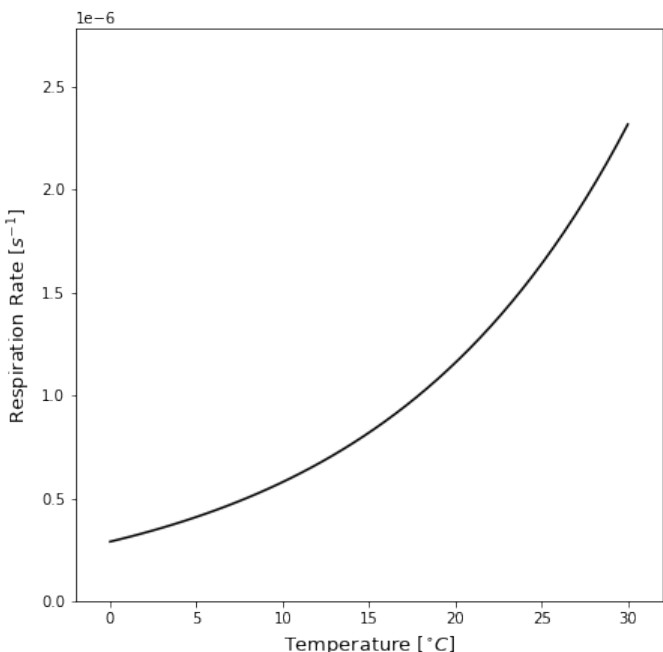

**Figure F1.** The algal rate of respiration $[s^{-1}]$ relative to the temperature of seawater $[^\circ C]$ as defined in Kooi et al. (2017) and in Eq. (8) in this study. Colder temperatures cause lower respiration rates and increasing the temperature exponentially increases the respiration rate. Since respiration is the dominant term below the MLD for almost all trajectories (Fig. 6), it defines the oscillatory behaviour of the particles.

## Appendix F: The rate of biofilm respiration relative to the seawater temperature

*Author contributions.* RF and DL co-developed the code. RF ran the simulations and analysis. DL lead the team and wrote the manuscript. MK and AK contributed advice on the Kooi et al. 2017 code modifications. VO contributed the code for the wind-mixing parametrisations. CL and LA-Z provided advice on the biological component of the model. AY contributed advice on the NEMO-MEDUSA data and modifications to the equations required. EvS contributed advice on OceanParcels simulations and is the TOPIOS team lead.

*Competing interests.* The authors declare that they have no conflict of interest.

*Acknowledgements.* RF, DL and EvS are part of the "Tracking Of Plastic In Our Seas" (TOPIOS) project, supported through funding from the European Research Council (ERC) under the European Union's Horizon 2020 research and innovation program (grant agreement no. 715386). AY was supported by the UK Natural Environment Research Council (NERC) CLASS project (NE/R015953/1). Simulations were carried out on the Dutch National e-Infrastructure with the support of SURF Cooperative (project no. 2019.034). CL and VO acknowledge support from the Swiss National Science Foundation under grant 174124. NOAA for grant NA17NOS9990024 awarded to LA-Z. We would like to thank Clément Vic from Ifremer for suggesting the background tidal-induced vertical mixing and providing the script to interpolate $K_z$ from de Lavergne et al (2020). The underpinning high-resolution NEMO-MEDUSA simulation was performed by Andrew Coward (NOC) using the ARCHER UK National Supercomputing Service (http://www.archer.ac.uk). We would also like to thank Hannah Kreczak, Andrew Baggaley and Andrew Willmott from the University of Newcastle for the discussions about the (Kooi et al., 2017) model and mixing.

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
