# Peer review of "Modeling submerged biofouled microplastics and their vertical trajectories"

_Biogeosciences, 2021_

## Author Comment (AC1)

**Responses to reviewer #1**

**General Comments**

This paper presents valuable contributions to the field of plastic modeling. Specifically, it helps address specific knowledge gaps in the vertical distribution of microplastics by focusing on biofouling, though it also includes other vertical mechanisms. This works goes even further to investigate global and seasonal variations that can impact biofilm growth vertical transport. Overall, the paper is clear and well-written, and I only recommend minor revisions before publication.

We thank reviewer #1 for the time spent reading and reviewing our paper, as well as their positive general comments. We appreciate that the reviewer believes that knowledge gaps are addressed with regards to the transport of ocean microplastic particles in the vertical dimension due to biofouling. We have responded to specific comments below.

**Specific Comments**

**The integration time step is stated to be 60 seconds around line 95. Did you examine the sensitivity of the model to this choice for time step? Is there some other justification?**

Yes, there is some justification so we have now added this in the text - we thank the reviewer for the question: L98-100 "We tested the sensitivity of our results using an integration time step of 30 seconds and the results did not change (though the simulation time was longer); with a longer integration time step (e.g. 1 hour) we lost some information for shorter oscillation frequencies (e.g. for 1 mm particles)."

**There is discussion on initial density around line 100 to justify the model choice of 920 kg/m3, but this only mentions comparison to particles with lower density (down to 30 kg/m3). Do you know what would happen for particles closer to the density of water but still positively buoyant such as HDPE?**

We thank the reviewer for this suggestion. We previously thought that since the results in Lobelle et al. 2021 show that the LDPE (920 kg/m3) and HDPE (940 kg/m3) sinking characteristics are very similar, it was enough justification for not running simulations for denser particles. Since we add vertical mixing in this study, however, the results could differ. We have now rerun the simulations with even denser particles (1020 kg/m3) representing rigid polyamide and the results are shown in the supplementary materials as Fig. C2.

---

## Author Comment (AC2)

**The manuscript presents a modeling study of the vertical distribution and trajectory of microplastics at three oceanic regions characterized by contrasted biological and physical properties. The study builds on the work in Lobelle et al. (2021), including improved parameterizations for vertical mixing and biofilm loss terms. The study is a good contribution to gain insight into the mechanisms driving the vertical transport of microplastics in the ocean. The manuscript is well written, with a clear structure. Therefore, I support the publication of this work.**

We would like to thank reviewer #2 for their time. Their expert feedback has been very useful in order to improve our manuscript. We have responded to the minor comments below individually.

**Some minor comments are given below:**

**The study is based on several model parameterizations and assumptions. The most important assumptions are discussed in Section 3.4 together with future potential developments. This section is quite pertinent, however, the model performance strongly depends on the calibration of a large number of parameters, a fact that may be further discussed or clarified :**

- **Parameters of mixing formulation: e.g. roughness scale, wave age, etc.**
- **Parameters of biofouling formulations: e.g. collision rate, growth rate, etc**
- **Would the combined use of a different plastic density and biofilm density change the results?**
- **How is the calibration/validation of NEMO-MEDUSA? Is it sensible to different parameterizations as well?**

We thank the reviewer for this comment. It is true that our model includes many parameters and though some have been tested for sensitivity purposes, it is almost impossible to test the full combination of a range of values for each parameter. Furthermore, most of the parameters are chosen following previous studies (either based on observations or models), which are referenced throughout the Methods section. We also ran some initial tests (not shown) to isolate the specific terms that showed a small effect on the results (e.g. plastic and biofilm density, in Appendix Figures B1 to D1). Finally, since this is a *process* study, we do not aim to get each parameter exactly right (since that is in fact impossible without validation through observations); we have therefore focused on defining the dependencies between terms and simulating the processes that dominate a particle's movement in the ocean over time and the vertical space.

To focus on the first bullet point, the roughness scale is based on Zhao et al. 2019 (and they state that the roughness scale is quite small at sea, where it mostly affects Kz at the surface, and then the rest of the profile is dominated by other terms). For the wave age, we have added the following specification: L153 "*Assuming a constant wave age for a fully developed wave state…*"

Parameters such as collision rate did not affect the results in our initial tests, however since we do not show these results, we agree that mentioning the need for further sensitivity analyses is appropriate in Section 3.4: L423-425 "*Lastly, further sensitivity analyses can be carried out regarding the collision rate, growth rate and other parameters that the model performance relies on, since analysing the full combination of all ranges of parameters was beyond the scope of this study.*"

Furthermore, we have added a supplementary figure (Fig. C2; see below) using a different plastic density (closer to the density of seawater; 1020 kg/m3 - representing rigid polyamide). The following has been added: L116-120 "*For the 1020 kg/m³ simulations in the NPSG (Fig. C2b and e), the majority of the larger particles mix completely to the base of the MLD (as opposed to 920 kg/m³ particles mostly staying close to the sea surface). The smaller 1020 kg/m³ particles on average resurface slower after being mixed down to 200 m in spring (as opposed to 920 kg/m³ particles that quickly resurface). Particles representing other sizes in other regions with a density of 30 and 1020 kg/m³ produce very similar results to the 920 kg/m³ particles.*" By simulating these three particle densities, the full range of floating ocean plastic is represented in our study.

[Figure]

Regarding the biofilm density, we have used 2 densities that are 'realistic' seeing as the one is based on the original Kooi et al. (2017) model (1388 kg/m3) and the other on observations (1170 kg/m3) by Amaral-Zettler et al. (2020). Any other densities have not been suggested in the literature.

Finally, regarding the validation of NEMO-MEDUSA, the global scale performance of the model has been evaluated in Yool et al. (2013) and Yool et al. (2021). While these comprehensive evaluations have taken place at low resolution, its performance at high resolution is traceable and very similar, although somewhat improved. Yool et al. (2015) includes a more limited evaluation at both low and high resolution. We have amended the manuscript to note these evaluations: L131-133 "*The biogeochemical performance of NEMO-MEDUSA has previously been extensively validated at low resolution in the studies of Yool et al. (2013) and Yool et al. (2021), with traceability at higher resolution demonstrated in Yool et al. (2015).*"

**Kooi's model seems to depend on water physical parameters such as temperature, salinity, viscosity. Did this work consider the seasonal and spatial variability of these parameters or are they just affecting the biofouling parameters in NEMO-MEDUSA? If so, how do they affect the spatial differences of the vertical distribution of particles between the three regions?**

Yes, the biofouling model in this work is dependent on physical seawater properties and includes their variability in time and space as provided by the NEMO-MEDUSA output. This is now described slightly clearer in Section 2.3 and 2.4: L180-181"*$rho_{sw}$ is the ambient seawater density [kg/m3] derived from NEMO-MEDUSA's temperature and salinity fields that vary in 3D time and space*". And L234-235 "*where T is the MEDUSA temperature field [C] that varies in 3D time and space (see Fig. F1 for the graphical relationship of the respiration rate and seawater temperature).*"

We have made the impact of this dependency slightly clearer at the beginning of Section 3.4 (Model assumptions and future model developments), for example, with the following sentences: L391-393 "*... the model relies strongly on the assumption that biofilm respiration depends only on temperature. After a particle is biofouled and sinks, continued respiration is the main mechanism for defouling, which in turn leads to the oscillation of the microplastic. Such behaviour is still theoretical and has never been experimentally observed.*" Furthermore, the length of oscillations are dependent on the temperature of water because respiration generally is the dominant term below the MLD (Fig. 5). This has also been added in the text: L385-387 "*It should also be noted that since respiration is the dominant process below the MLD in general, and respiration is dependent on temperature (Eq. 8), the oscillatory behaviour of particles is dependent on the surrounding water temperature (Appendix F).*"

The vertical velocity of the particle ($w_s$) is dependent on the surrounding seawater density, where the regionally averaged MLD is an indication of the stratification of the water. This has been made clearer in the text as follows: L275-278 "*In general, in regions with more mixing (and less stratification), a particle that is moving vertically will be less affected by sudden changes in density and can sink deeper (for example, in the SO), whereas in regions with less mixing (more stratification), the opposite occurs and particles tend to sink to shallower depths (for example, in the NPSG)*".

**Unless I am mistaken, the discussion of results from Appendix D is quite light. Given that the oscillatory behavior of microplastics has not been observed yet, it may be interesting to elaborate on this scenario that may also represent a "realistic" situation.**

We agree with the reviewer that the results from Appendix D are limited. We must mention that although this could seem like a more 'realistic' scenario, without any observations regarding what happens to biofilms below the EZD, this is just one potential model. We do not wish to focus too much on this model since we are simply presenting one alternative (or adaptation) to the Kooi model and further work should be done to expand and test it further.

We have added a sentence, however, to introduce the model in the Methods firstly: L242-244 "*These biofilm gain and loss terms result in an oscillatory behaviour of the particles due to the biofilm's gain causing an increase in overall density and sinking followed by the biofilm's loss and decrease in density which leads to rising. We also propose an alternative scenario where biofilm cells remain attached in the dark (see full description in Appendix D).*"

We have also added a sentence in the Results section: L321-324 "*Across all regions, the sensitivity analysis simulating the denser algal cell wall that remains attached after the biofilm is dead shows that oscillations still occur (Fig. D1). For the smaller size class, particles can reach deeper depths and longer oscillations, however the larger size class remains unchanged. Since this phenomenon has never been experimentally observed, we suggest one alternative approach for the biofilm dynamics.*"

**Line 25: A new study published in Science (Weiss et al., 2021) reformulates the calculation of plastic fluxes and shows that there may not be this huge amount of "missing plastic". Weiss, L., Ludwig, W., Heussner, S., Canals, M., Ghiglione, J.F., Estournel, C., Constant, M., Kerhervé, P. (2021). The missing ocean plastic sink: Gone with the rivers, Science, 373 (6550), 107-111**

It is true Weiss et al. (2021) suggest that the Jambeck et al. (2015) study overestimates the amount of plastic entering the oceans. However, regardless of the plastic influx into the oceans, the simulations in Onink et al. (2021) would still produce the same results regarding the fraction that ends up on beaches or coastal waters. We have included this in the introduction as follows: L28-L30 "*Although a recent study by Weiss et al. (2021) suggests that Jambeck et al. (2015) overestimates plastic fluxes from rivers to oceans by two to three orders of magnitude, this would not affect the fraction of the total that ends up close to the coasts from Onink et al. (2021), for example. Following these approximations, around 20-30 % of ocean plastic debris is unaccounted for and could either be in the water column or on the seafloor.*"

**References**

Amaral-Zettler, L. A., Zettler, E. R., and Mincer, T. J.: Ecology of the plastisphere, Nature Reviews Microbiology, https://doi.org/10.1038/s41579-019-0308-0, 2020.

Kooi, M., Nes, E. H. V., Scheffer, M., and Koelmans, A. A.: Ups and Downs in the Ocean: Effects of Biofouling on Vertical Transport of Microplastics, Environmental Science and Technology, 51, 7963–7971, https://doi.org/10.1021/acs.est.6b04702, 2017.

Onink, V., Jongedijk, C. E., Hoffman, M. J., van Sebille, E., and Laufkötter, C.: Global simulations of marine plastic transport show plastic trapping in coastal zones, Environmental Research Letters, 16, https://doi.org/10.1088/1748-9326/abecbd, 2021.

Yool, A., Popova, E. E., and Anderson, T. R.: MEDUSA-2.0: An intermediate complexity biogeochemical model of the marine carbon cycle for climate change and ocean acidification studies, Geoscientific Model Development, 6, 1767–1811, https://doi.org/10.5194/gmd-6-1767-2013, 2013.

Yool, A., Popova, E. E., and Coward, A. C.: Future change in ocean productivity: Is the Arctic the new Atlantic, Journal of Geophysical Research: Oceans, 120, 7771–7790, https://doi.org/10.1002/2015JC011167, 2015.

Yool, A., Palmiéri, J., Jones, C. G., Mora, L. D., Kuhlbrodt, T., Popova, E. E., Nurser, A. J. G., Hirschi, J., Blaker, A. T., Coward, A. C., Blockley, E. W., and Sellar, A. A.: Evaluating the physical and biogeochemical state of the global ocean component of UKESM1 in CMIP6 Historical simulations, Geoscientific Model Development, https://doi.org/10.5194/gmd-14-3437-2021, 2021.

Zhao, D. and Li, M.: Dependence of wind stress across an air–sea interface on wave states, Journal of Oceanography, 75, 207–223, https://doi.org/10.1007/s10872-018-0494-9, 2019.

---

## Author Comment (AC3)

**Responses to reviewer #3**

**This is a very interesting paper building on previous work by some of the authors, notably, Kooi et al. (2017) and Lobelle et al. (2021). The authors investigate the dynamics of submerged plastic particles in response to vertical advection, diffusion and biofouling using for that a well-established Lagrangean model of marine plastic transport. Main results of this study are that plastic particles still undergo oscillations, similar to those discussed by Kooi et al. (2017) and Lobelle et al. (2021), when the effects of vertical advection and diffusion are taken into account. Subsurface maximum concentrations of plastics can be created in low turbulence areas when downward settling is approximately balance by ambient upwelling. Strong mixing can draw plastic particles down to hundreds of metres below the surface, for example in the Southern Ocean.**
**The paper is well written, well referenced for the most part, and the arguments put forward to explain the model results are generally rigorous and persuasive. I have nevertheless quite a few comments and remarks to make, and I have some objections regarding aspects of the methodology and interpretation of the results. This makes me recommend major revisions. In the following, my comments are organised by line number.**

We would first like to thank the reviewer for their general comments as well as their extensive specific comments below. We feel that with the changes to the manuscript that were suggested by this reviewer, our paper has vastly improved. Please find the responses to the comments below:

**Ll. 41-45.**
We have split this comment into 3 parts to make it easier to answer:

**Vertical mixing within the mixed layer is driven both by wind stirring and buoyancy losses. While the Kulkuka et al. (2012) parameterisation accounts only for wind mixing, the mixing that takes place in the NEMO-MEDUSA and that accounts in part, for the vertical distribution of phytoplankton and primary production the authors use in their simulations, takes into account both wind stirring and instabilities driven by buoyancy losses. Similarly, vertical mixing of plastic particles in the mixed layer is caused by both turbulent mechanisms, not just wind-induced mixing.**

That is correct, we thank the reviewer for pointing this out. We have edited the text to include this: see L45-46 "*Model studies simulating the effects of turbulent mixing driven by wind and buoyancy loss demonstrate that…*"

**In a similarly vein, mixing in the ocean interior stems from many causes, one of them being the dissipation of tidal energy on the shelves (this mixing will often overlap with mixing originating from surface buoyancy and momentum fluxes) and in the deep ocean over rough topography. But dissipation of tidal energy is not the sole source of turbulent kinetic energy in the deep ocean and, away from boundaries and rough bathymetry, not even the most important. So, it is not entirely appropriate to refer to ocean interior mixing as tidal mixing.**

Our study focuses on open-ocean (not shelf seas) dynamics, where mixing in the interior is at first order fuelled by the breaking of internal tides, and not only over rough seafloor topography. This was demonstrated in de Lavergne et al. (2020) that compared a

parametrisation for internal tide driven mixing with in situ measurements of mixing, which hence encapsulate all mixing processes. The two datasets compare remarkably well, see Figures 8 and 10 in de Lavergne et al. (2020), highlighting that tides are indeed the major contributor to mixing. Quote from de Lavergne et al. (2020) : "*The comparison also suggests that, in the ocean interior, internal tides are the principal energy source for mixing and are responsible for the main large-scale patterns of mixing. This inference is consistent with the large power input to internal tides (~1 TW) and with the relative temporal stability of basin-scale dissipation patterns (Ferron et al., 2016; Kunze, 2017).*"

We have added a sentence to explicitly mention this in the introduction, see L49-50 "*Furthermore, de Lavergne et al. (2020) have shown that mixing in the interior is at first order governed by mixing driven by internal tides.*"

**Finally, Mountford and Morales Maqueda (2019, 2020) have also demonstrated the importance of diapycnal mixing through the water column in controlling the penetration of buoyant microplastics in the water column. Please acknowledge.**

We agree with this statement, and believe that we have been misunderstood. Tidally-driven mixing is by essence diapycnal, hence we have also made this explicit in the text, see L50-51"*We therefore include tidally-induced diapycnal mixing in this study…*" and well as in the Methods on L135-136: "*There are three components that make up the physical dynamics included in this study: vertical advection (from NEMO-MEDUSA), computed wind-induced vertical mixing and computed tidally-induced diapycnal vertical mixing.*"

**Ll. 52-54. That these oscillations may exist is entirely plausible but, to my knowledge, they have never been identified in the field. One of the beauties of modelling is that it allows us to explore and predict phenomena that have not yet been observed. Given the growing amount of literature on the subject of these plastic oscillations, it is becoming urgent to come up with ideas on how to gather observational evidence of the process to ascertain whether it is actually taking place in the real world.**

Yes, the reviewer is correct that the oscillations have never been observed and validating these models with in situ data is becoming very necessary. We believe it is beyond our expertise to suggest exactly how the observational evidence should be conducted since we are unsure of the feasibility of such open ocean experiments on particles smaller than 1mm. We have, however, made it clearer that collecting such samples is urgent for future developments in Section 3.4.: L442-444 "*Overcoming some of the logistical challenges to measure and monitor the 3D movement of plastic (specifically smaller than 1 mm) in the open ocean is becoming urgent in order to validate that: (1) biofouled particles oscillate (since this has never been observed)...*"

**Ll 95-96. The validity of Stokes' law does (not) depend on the radius of the particles but on the Reynolds number. Please, correct and provide the correct reason, which much be that buoyant particles with a larger radius will shift the motion regime to one of high Reynolds numbers for with Stokes' law needs to be corrected.**

This is a good point by the reviewer. Indeed, the use of 'Stokes motion equation' here may have been confusing. We have now removed the link to Stokes in the text, and only state that the largest size is equal to that used in Lobelle et al (2021). Concerning the

validity: Figure 2 (below) of De La Fuente et al (2021) suggests that for 1 mm particles, the Maxey-Riley equation is valid when $w_s < 0.02$ m/s. This is indeed the case for the settling velocity in most of our simulations (green bars in Figures A1, A2 and A3), especially below the mixed layer. We therefore prefer to keep these simulations in our manuscript, but have added this caveat to L433-436 "*The Kooi et al. (2017) model also places limits on the maximum size of the particles, as De La Fuente et al. (2021) show that for particles of size 1 mm, the Maxey-Riley-model is only valid for sinking velocities below 2 cm/s; a condition that is just about met below the mixed layer in our simulations (green bars in Figs. A1-A3).*"

[Figure]

**Figure 2.** Settling velocities and particle sizes for which Eq. (3) holds. Kolmogorov scale is represented by the red line and $Re_p = 1$ with a black line, which bound the area of validity. Blue curve corresponds to $v_s = v_s(\beta, a)$ for $\beta = 0.8$, the upper bound to $v_s$ for typical microplastic densities. Dark shading denotes the plastic particle sizes and corresponding settling velocities for which application of Eq. (3) is valid.

**LI 100-105. Are you sure that the use of particles as light as 30 kg m⁻³ does not push the system to terminal Reynolds number much smaller than 1, outside the permissible Stokes regime?**

Again, this is a good point by the reviewer. However, since we start all particles at the surface, their sinking (and thus the effect of Stokes law) will only start after their total density (plastic + biofilm) is larger than that of sea water. In that sense, it does not matter very much what the initial density of the plastic itself is (as also confirmed by Fig C1).

**LI 115-117. One-dimensionality is a dangerous assumption here. Water is nearly incompressible, and incompressible fluid motion is never one-dimensional if the divergence of the flow in any one direction is non-zero. This must be, in general, the case in your simulations. For example, an annual mean upward vertical velocity of 10⁻⁹ m s⁻¹ (Fig. A1) at the base of an equatorial column of water extending over a 1∘1∘ area and 1000 m deep would cause a horizontal divergent current of about 1 cm s⁻¹, which means that any particle of plastic originally within the 1∘1∘ would have left the area in about one hundred days.**

We appreciate this comment from the reviewer. Though we understand and agree that water is nearly incompressible and we should ideally account for that, we still believe that using only vertical velocities is the added value of this study - we will explain why. We have selected three regions with contrasting oceanic biological and physical conditions and properties. We therefore want our simulations to represent a clear regional signal over the period of a year, which consequently allows us to analyse

seasonality. When, for example, including horizontal physical processes in the Equatorial Pacific, the particles will quickly move poleward, which prevents us from being able to state that in the Equatorial Pacific, a biofouled particle behaves a certain way.

To accommodate the reviewer, we have now also run some tests with 3D advection included, and confirmed that this does not drastically change the results (see figures below and added in Appendix E in the manuscript). The maximum depth is similar and the seasonality also shows similar patterns.

We have added this sentence in the Methods: L126-129 "*As the aim of our study is to analyse the regional signal, we run all simulations in the vertical dimension only, however we have also tested the sensitivity of our results to 3D advection (Appendix E), where the horizontal velocities from NEMO-MEDUSA are also used for these simulations.*" Also, in the results we have added the following: L259-261 "*We have also tested the effects of 3D advection (Appendix E) and we demonstrate that even though particles can travel for thousands of kilometers (Fig. E1) after a few months (e.g. in the EqPac), the results are not largely impacted (Figs. E2-E4).*"

[Figure]

**Figure E2.** As in Fig 2 but for particles only in the EqPac and running for 180 days. The left 2 figures are without advection and the right 2 figures are with advection. This means that (a) and (c) are identical to Fig 2a and Fig 2d, however, including the 3-month spin-up time and running for fewer days (not a full year).

[Figure]

**Figure E3.** As in Fig E2 but for particles only in the NPSG. This means that (a) and (c) are identical to Fig 2b and Fig 2e, however, including the 3-month spin-up time and running for fewer days (not a full year).

[Figure]

**Figure E4.** As in Fig E2 but for particles only in the SO. This means that (a) and (c) are identical to Fig 2c and Fig 2f, however, including the 3-month spin-up time and running for fewer days (not a full year).

**Ll 118-120. This is a rather bizarre statement. It is not clear to what is meant by "very low-resolution of turbulence". In ocean models, the resolution of diapycnal turbulence is as high or as low as that of any other vertical or quasi-vertical process. If you discard the turbulence calculated by the NEMO-MEDUSA model because they are allegedly of too low resolution, it is not clear to me why would you think acceptable to use the vertical velocities produced by the same model and which are computed on the same low resolution, especially since the dynamics of turbulence and vertical advection are intimately related. The formulation of turbulent vertical diffusivities in NEMO-MEDUSA is based in the TKE scheme of Gaspar et al. (1990) which is one of the best physically founded and conceptually appealing formulation of these processes for ocean modelling. Like the KPP formulation, the TKE scheme is capable of representing non-uniform turbulent profiles in the simulated mixed layer as long as the said mixed layer is resolved well enough by the model. This is not to denigrate the author's choice of the KPP parameterisation to calculate diffusivities in their model. I presume this is because they do not have access to the diffusivities produced by the NEMO-MEDUSA model. This is fine as a justification for your approach. But then, it suffices to state so. There is no need to advance questionable arguments about the resolution of turbulence in NEMO-MEDUSA.**

We thank the reviewer for this comment and we agree that stating that the resolution of turbulence in NEMO-MEDUSA is very low is not correct and not necessary. As the reviewer expected, we did not have access to the diffusivity files from NEMO-MEDUSA so we have stated this in the text and removed any mention of resolution, see L140-144 "*Wind-driven turbulence can play an important role in the vertical concentration profiles of buoyant particles (Kukulka et al., 2012) and therefore its inclusion is one of the novelties of this study (Table 1). Since we do not have access to the diffusivity profiles from NEMO-MEDUSA, we follow the approach from Onink et al. (subm.) to model turbulent stochastic transport in the surface mixed layer using a Markov-0 random walk model.*"

**L 125 u_star is called the friction, not frictional, velocity (see also L 130). It does not correspond to any velocity of the seawater at the surface or anywhere else. It is simply a way of expressing a stress, in this case the surface wind stress, in terms of a velocity.**

We have changed all instances of "frictional" to "friction".

**L 127. Ignoring the impact of Langmuir enhanced mixing in the Southern Ocean might be problematic but perhaps acceptable here for simplicity.**

We have now included this as follows: L150-152 "*Note that neglecting Langmuir enhanced mixing in the Southern Ocean might not be realistic (Li et al. 2016), however for simplicity and standardisation, we keep this assumption constant in all regions.*"

**Ll 133-135. This statement is also rather obscure. The non-locality of the KPP scheme has nothing to do with its (inexistent) dependence on the wind stress at different locations. It refers to the fact that, in cases of unstable forcing, turbulent fluxes are decomposed into a local term that is proportional to the vertical gradient of the property being fluxed and a non-local term that conveys information of the surface property flux down through the boundary layer. There**

**is, however, nothing wrong with neglecting this nonlocal term, again, for simplicity.**

We agree with the reviewer that this sentence was not very precise so we have simplified it to: L156-160 "*We use a local form of the KPP profile, where we neglect non-local terms for simplicity.*"

**L 143. It is Delta t, not delta t**

We thank the reviewer for noticing this, the change has now been made on L167.

**LI 146-148. Ocean interior mixing is driven by tidal processes as well as by processes other than tides. Instead of an extraneous set of diffusivities, it might have been desirable to use the diffusivity output from NEMO-MEDUSA, which includes all relevant sources of turbulent forcing and, very importantly, is physically compatible with the vertical velocities you are using. I appreciate this might not be possible, though. In any case, could the authors please show the vertical diffusivity profiles (in annual mean, say) they are using in the simulations for the three sites?**

We thank the reviewer for this question - we believe that the answer is exactly the same as 5 comments prior (regarding using the KPP profiles methodology). We simply did not have the diffusivity profiles from NEMO-MEDUSA available. Furthermore, we would like to refer the reviewer to Fig. 1 in the manuscript, where we believe the requested profiles are already shown (see below). The blue lines (top x-axis) represent the annually averaged vertical diffusivity profiles for the three regions:

[Figure]

**LI 164-166 Please expand. In which way is the assumption of diatom-dominated biofilms better than one in which both diatoms and non-diatoms are combined?**

Linda Amaral-Zettler, the co-author with expertise in the biofilm's biological community (plastisphere) suggests that we do not use non-diatoms based on their observational

study. This has now been made clearer in the text: L190-193 "*...Amaral-Zettler et al. (2020) did not observe the species of phytoplankton that make up the bulk of the non-diatoms in their observational study of biofilms. Therefore we choose to limit the species to diatoms, which we know to be found on ocean plastic.*"

**LI 236-238. As indicated above, other authors have, however, demonstrated the importance of advection and diffusion processes, both diapycnal (~vertical) and isopycnal (~horizontal), for the distribution of microplastics in the ocean.**

Yes, this is true - and this is why we decided to test the sensitivity of our results to the addition of 3D advection, thanks to the suggestion from the reviewer. We would like to refer the reviewer to the sentences (already referenced above) regarding the fact that the results remain largely similar. This has been mentioned here: L259-261"*We have also tested the effects of 3D advection (Appendix E) and we demonstrate that even though particles can travel for thousands of kilometers (Fig. E1) after a few months (e.g. in the EqPac), the results are not largely impacted (Figs. E2-E4).*"

**L 265. I find this phrase awkward, could you please rephrase?**

We have rephrased this to: L304-306 "*One possible explanation for this is an equilibrium between the biological and physical processes that can cause upwards or downwards movement of a particle.*"

**L 288. I am not sure this is the most accurate way of describing what is happening. The particles in the mixed layer are being strongly mixed (passively), rather than moving "passively with the flow".**

We agree with the reviewer and this section now reads: L332-335 "*Throughout all the simulations, two distinct horizontal layers are formed, one above the MLD, where the ambient vertical velocities dominate (and particles are strongly mixed, passively) and one below the MLD, where the particles' settling velocity dominates (and particles move actively, relative to the flow).*"

**Figure 4 caption. What, pray tell, is Wmixing? Is this given by (3). If so, use a consistent notation.**

We thank the reviewer for this suggestion since it is true that we had two different notations in the text and the figure for the same term. We have now changed all vertical velocity terms so that they start with "$w$". Therefore, vertical velocities related to turbulent mixing as indeed described by Eq. (3) are now "$w_m$", vertical advection is now "$w_a$" and vertical settling is now "$w_s$". In Figure 4, the colorbar remains $w_{mixing}$ etc. and is explained in the caption so that it is easier for the reader looking at the figures before reading the text what is being shown. Caption for Fig. 4: "*The ratio between a particle's absolute settling velocity ($w_{settling}$ or $w_s$ from Eq. (5)) and the absolute ambient vertical velocity; mixing plus advection ($w_{advection} + w_{mixing}$ or $w_m$ from Eq. (3))....*"

**L312 A potential reason? Please, look at the data of primary production you are using and confirm or disprove your hypothesis. There is no reason here to advance a guess when the actual explanation is at hand.**

We investigated the primary productivity from the NEMO-MEDUSA data to see whether we could explain the subsurface dominance in biofilm growth that occurs between 150

and 250 m during the spring months in the NPSG. Below is a plot of the monthly-averaged primary productivity rate profile:

[Figure]

There is no subsurface maximum primary production from February to April between 150-250 meters so this theory can be disproved. We then looked at the nutrient concentration profile data from NEMO-MEDUSA and found that dissolved inorganic nitrate increases linearly with depth:

[Figure]

Therefore, even though there is very low primary productivity, the presence of the biofilm allows for the algae to grow. We have now added the following sentences in the Results: L360-365 " …*To explain this, we looked at the profiles of dissolved inorganic nitrate from NEMO-MEDUSA (not shown), where the surface is nutrient-depleted but below 100 m, concentrations increase linearly. This means that between 150 and 250 m there is sufficient nitrate for the biofilm to grow, but light is limiting and at the surface the opposite is true (nutrients are limiting but there is sufficient light). This results in* Ggrow *having a similar order of magnitude at the surface and 150-250 m.*"

**L 325 Equation (5) does not readily show that larger particles have larger settling velocities than smaller ones, although your statement is correct as long as the densities of the particles are the same.**

This is true. We have therefore instead decided to change the reference to our supplementary figures (A1-A3) and a figure in the Kooi et al. (2017) paper (see below). L375-376 now reads: "*Larger particles have a higher sinking velocity than smaller particles (see green bars in Figs. A1-A3 and Fig. 3 in Kooi et al. 2017)*"

[Figure]

**L 343 heterotrophs.**

This typo has been fixed (from heterotophs to heterotrophs on L403).

**L409 Lresp is repeated twice.**

We also thank the reviewer for spotting this mistake. The text has been changed to: L474-475 "We simulate that the living algal loss terms (*Lgraze* in Eq. (7), *Lresp* in Eq. (8) and *Lnonlin* in Eq. (9))..."

**References**
Amaral-Zettler, L. A., Zettler, E. R., and Mincer, T. J.: Ecology of the plastisphere, Nature Reviews Microbiology, 490 https://doi.org/10.1038/s41579-019-0308-0, 2020.

de la Fuente, R., Drótos, G., Hernández-García, E., López, C., and Van, E.: Sinking microplastics in the water column: simulations in the Mediterranean Sea, Ocean Science, pp. 1–32, 2020.

de Lavergne, C., Vic, C., Madec, G., Roquet, F., Waterhouse, A. F., Whalen, C. B., Cuypers, Y., Bouruet-Aubertot, P., Ferron, B., and Hibiya, T.: A Parameterization of Local and Remote Tidal Mixing, Journal of Advances in Modeling Earth Systems, 12, https://doi.org/10.1029/2020MS002065, 2020.

Kooi, M., Nes, E. H. V., Scheffer, M., and Koelmans, A. A.: Ups and Downs in the Ocean: Effects of Biofouling on Vertical Transport of Microplastics, Environmental Science and Technology, 51, 7963–7971, https://doi.org/10.1021/acs.est.6b04702, 2017.

Kukulka, T., Proskurowski, G., Morét-Ferguson, S., Meyer, D. W., and Law, K. L.: The effect of wind mixing on the vertical distribution of buoyant plastic debris, Geophysical Research Letters, 39, https://doi.org/10.1029/2012GL051116, 2012.

Li, D., Liu, K., Li, C., Peng, G., Andrady, A. L., Wu, T., Zhang, Z., Wang, X., Song, Z., Zong, C., Zhang, F., Wei, N., Bai, M., Zhu, L., Xu, J., Wu, H., Wang, L., Chang, S., and Zhu, W.: Profiling the Vertical Transport of Microplastics in the West Pacific Ocean and the East Indian Ocean with a Novel in Situ Filtration Technique, Environmental Science and Technology, 54, 12 979–12 988, https://doi.org/10.1021/acs.est.0c02374, 2020.

Lobelle, D., Kooi, M., Koelmans, A. A., Laufkötter, C., Jongedijk, C. E., Kehl, C., and van Sebille, E.: Global modeled sinking characteristics of biofouled microplastic, Journal of Geophysical Research: Oceans, pp. 1–15, https://doi.org/10.1029/2020jc017098, 2021.

Onink, V., van Sebille, E., and Laufkötter, C.: Empirical Lagrangian parametrization for wind-driven mixing of buoyant particles at the ocean 580 surface (submitted), pp. 1–19.

---

## Author Response (AR2)

**Final Author Response**
**9 March 2021**

Although the paper was accepted to be published as is, we decided to include the 2 citations that were mentioned in the Report by Reviewer #3 and the editor. These had not been intentionally omitted and are a valuable contribution to our paper:

Lines 47-49: Furthermore, \citet{deLavergne2020} have shown that mixing in the interior is at first order governed by mixing driven by internal tides. **Previous work by \citet{Mountford2019,Mountford2021} has also shown the importance of interior diapycnal mixing for the dispersal of plastic in the ocean**. We therefore include diapycnal mixing (tidally-induced) in this study both to test whether it can impact near-surface particle displacement as well as to provide full-depth mixing dynamics (and not solely within the mixed layer).